# The Global Fire Atlas of individual fire size, duration, speed, and direction

Niels Andela[1,2], Douglas C. Morton[1], Louis Giglio[3], Ronan Paugam[4], Yang Chen[2], Stijn Hantson[2],
Guido R. van der Werf[5], and James T. Randerson[2]

[1]Biospheric Sciences Laboratory, NASA Goddard Space Flight Center, Greenbelt, MD 20771, USA
[2]Department of Earth System Science, University of California, Irvine, CA 92697, USA
[3]Department of Geographical Sciences, University of Maryland, College Park, MD 20742, USA
[4]Centre Européen de Recherche et de Formation Avancée en Calcul Scientifique, URA1875, CNRS,
Toulouse, France.
[5]Faculty of Science, Vrije Universiteit Amsterdam, Amsterdam, Netherlands

*Correspondence to:* Niels Andela (niels.andela@nasa.gov)

**Abstract.** Natural and human-ignited fires affect all major biomes, altering ecosystem structure, biogeochemical cycles, and atmospheric composition. Satellite observations provide global data on spatiotemporal patterns of biomass burning and evidence for rapid changes in global fire activity in response to land management and climate. Satellite imagery also provides detailed information on the daily or sub-daily position of fires that can be used to understand the dynamics of individual fires. The Global Fire Atlas is a new global dataset that tracks the dynamics of individual fires to determine the timing and location of ignitions and fire size, duration, daily expansion, fire line length, speed, and direction of spread. Here we present the underlying methodology and Global Fire Atlas results for 2003-2016 derived from daily moderate resolution (500 m) Collection 6 MCD64A1 burned area data. The algorithm identified 13.3 million individual fires over the study period, and estimated fire perimeters were in good agreement with independent data for the continental United States. A small number of large fires dominated sparsely populated arid and boreal ecosystems, while burned area in agricultural and other human-dominated landscapes was driven by high ignition densities that resulted in numerous smaller fires. Long-duration fires in the boreal regions and natural landscapes in the humid tropics suggest that fire-season length exerts a strong control on fire size and total burned area in these areas. In arid ecosystems with low fuel densities, high fire spread rates resulted in large, short-duration fires that quickly consumed available fuels. Importantly, multi-day fires contributed the majority of burned area in all biomass burning regions. A first analysis of the largest, longest, and fastest fires that occurred around the world revealed coherent regional patterns of extreme fires driven by large-scale climate forcing. Global Fire Atlas data are publicly available through www.globalfiredata.org, and individual fire information and summary data products provide new information for benchmarking fire models within ecosystem and Earth system models, understanding vegetation-fire feedbacks, improving global emissions estimates, and characterizing the changing role of fire in the Earth system.

# 1 Introduction

Worldwide, fires burn an area about the size of the European Union every year (423 Mha yr$^{-1}$; Giglio et al., 2018). The majority of burned area occurs in grasslands and savannas where fires maintain open landscapes
by reducing shrub and tree cover (Scholes and Archer, 1997; Abreu et al., 2017). However, all major biomes burn. Climate controls global patterns of fire activity by driving vegetation productivity and fuel build up as well as fuel moisture (Bowman et al., 2009). Humans are the dominant source of ignitions in most flammable ecosystems, but human activities also reduce fire sizes through landscape fragmentation and fire suppression (Archibald et al., 2012; Taylor et al., 2016; Balch et al., 2017).

Over the past 18 years, socio-economic development and corresponding changes in human land use have considerably reduced fire activity in fire-dependent grasslands and savannas worldwide (Andela et al., 2017). At the same time warming climate has dried fuels and has increased the length of fire seasons across the globe (Jolly et al., 2015), which is particularly important in forested ecosystems with abundant fuels
(e.g., Kasischke and Turetsky, 2006; Aragão et al., 2018). Fire activity increases non-linearly in response to drought conditions in populated areas of the humid tropics (Alencar et al., 2011; Field et al., 2016), resulting in large scale degradation of tropical ecosystems (van der Werf et al., 2008; Morton et al., 2013b; Brando et al., 2014), and extensive periods of poor air quality (Johnston et al., 2012; Lelieveld et al., 2015; Koplitz et al., 2016). Moreover, increasing population densities in highly flammable biomes also amplify
the socio-economic impacts of wildfires related to air quality or damage to houses and infrastructure (Moritz et al., 2014; Knorr et al., 2016). Despite the importance of understanding changing global fire regimes for ecosystem services, human well-being, climate, and conservation, our current understanding of changing global fire regimes is limited because existing satellite data products detect actively burning pixels or burned area, but not individual fires and their behavior.

Frequent observations from moderate-resolution, polar-orbiting satellites may provide information on individual fire behavior in addition to estimates of total burned area. Several recent studies have shown that fire-affected pixels can be separated into clusters based on spatial and temporal proximity. This information can be used to study the number and size distributions of individual fires (Archibald and Roy, 2009; Hantson
et al., 2015; Oom et al., 2016), fire shapes (Nogueira et al., 2017; Laurent et al., 2018), and the location of ignition points (Benali et al., 2016; Fusco et al., 2016). One limitation of fire clustering algorithms that rely on spatial and temporal proximity of fire pixels is the inability to separate individual fires within large burn patches that contain multiple ignition points, a frequent phenomenon in grassland biomes. To address the possibility of multiple ignition points, other algorithms have specifically tracked the spread of individual
fires in time and space, with demonstrated improvements for isolating ignition points and constraining final fire perimeters (Frantz et al., 2016; Andela et al., 2017). In addition to the size and ignition points of individual fires, other studies used daily or sub-daily detections of fire activity to track growth dynamics of fires (Loboda and Csiszar, 2007; Coen and Schroeder, 2013; Veraverbeke et al., 2014; Sá et al., 2017). Together, these studies highlight the strengths and limitations of using daily or sub-daily satellite imagery
to derive information about individual fires and their behavior over time.

Here we present the Global Fire Atlas of individual fires based on a new methodology to identify the location and timing of fire ignitions and estimate fire size, duration, daily expansion, fire line, speed, and direction of spread. The Global Fire Atlas is derived from the Moderate Resolution Imaging
Spectroradiometer (MODIS) collection 6 burned area dataset (Giglio et al., 2018), which includes an estimated day of burn data layer at 500 m resolution. Individual fire data were generated starting in 2003, when combined data from the Terra and Aqua satellites provide greater burn date certainty. The algorithm for the Global Fire Atlas tracks the daily progression of individual fires at 500 m resolution to produce a set of metrics on individual fire behavior in standard raster and vector data formats. Together, these Global
Fire Atlas data layers provide an unprecedented look at global fire behavior and changes in fire dynamics

during 2003-2016. The data are freely available at http://www.globalfiredata.org, and new years will be added to the dataset following the availability of global burned area data.

## 2 Data and Methods


Here we developed a method to isolate individual fires from daily moderate resolution burned area data. The approach used two filters to account for uncertainties in the day of burn in order to map the location and timing of fire ignitions and the extent and duration of individual fires (Fig. 1). Subsequently, we tracked the growth dynamics of each individual fire to estimate the daily expansion, daily fire line, speed and

direction of spread. Based on the Global Fire Atlas algorithm, burned area was broken down into seven fire characteristics in three steps (Fig. 1b). First, burned area was described as the product of ignitions and individual fire sizes. Second, fire size was further separated into fire duration and a daily expansion component. Third, the daily fire expansion was subdivided into fire speed, the length of the fire line, and the direction of spread. The Global Fire Atlas algorithm can be applied to any moderate resolution daily

global burned area product, and the quality of the resulting dataset depends both on the Fire Atlas algorithm as well as the underlying burned area product. Here we applied the algorithm to the MCD64A1 collection 6 burned area dataset (Giglio et al., 2018) and the minimum detected fire size is therefore one MODIS pixel (approximately 21 ha). Several studies have shown that the MCD64A1 collection 6 burned area product provides a considerable improvement compared to previous generation of moderate resolution global

burned area products (Giglio et al., 2018; Humber et al., 2018; Rodrigues et al., 2019). We also present a preliminary accuracy assessment of the higher order Global Fire Atlas products using independent fire perimeter data for the continental US and active fire detections to assess estimated fire duration and the temporal accuracy of individual fire dynamics.

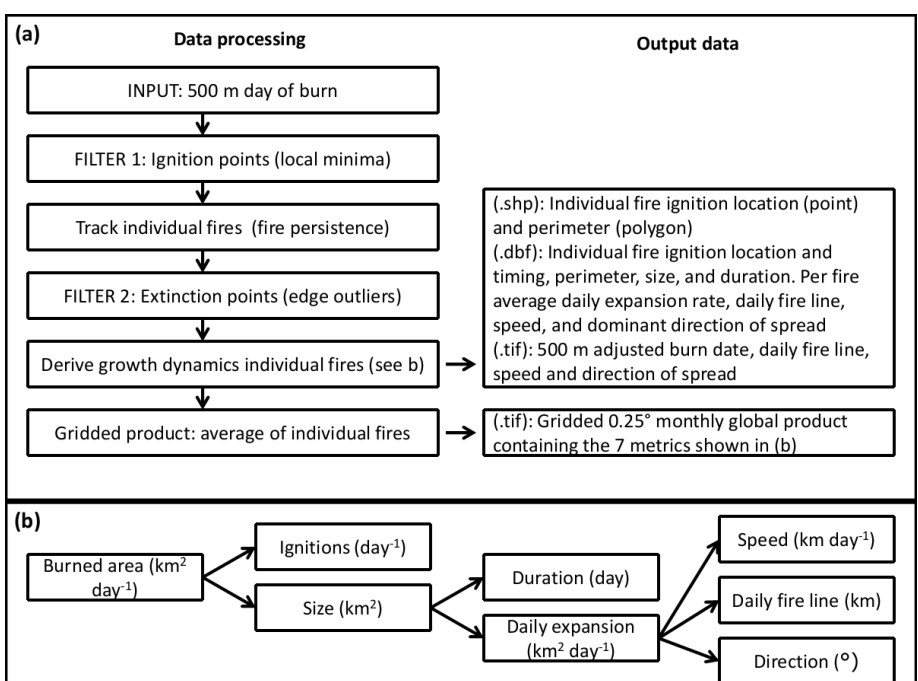


**Figure 1: Flow chart showing the data-processing steps and resulting products.** (a) The Global Fire Atlas algorithm tracks individual fires and their day-to-day behavior based on the MCD64A1 collection 6 500 m daily burned area product starting in 2003. (b) Decomposition of burned area into seven different components of the fire regime in the Global Fire Atlas. The output includes two annual shapefile layers

(.shp) of ignition location and individual fire perimeters with corresponding database files (.dbf) providing summary information for each individual fire, including the seven key characteristics. In addition, four per

fire year global raster maps on the 500 m sinusoidal MODIS grid (.tif) provide details on the day-to-day fire behavior. Finally, data are summarized in a monthly 0.25° gridded product based on average values of individual fires. Global Fire Atlas data-layers are described in more detail in Table A1.


## 2.1 Individual fires: ignitions, size, perimeter and duration

Large burn patches are often made up of multiple individual fires that may burn simultaneously or at different points in time during the fire season, particularly in frequently burning grasslands and savannas
with a high density of ignitions from human activity. Separating large clusters of burned area into individual fires is therefore critical to understand the fire regime in human-dominated landscapes. To isolate individual fires, clusters of adjacent burned area for a given fire season (12 months centered on the month of maximum burned area) were subdivided into individual fires based on the spatial structure of estimated burn dates in the MCD64A1 burned area product. Although we allow individual fires to burn from one fire season into
the next, we processed the data on a per-fire-season basis in each 10° x 10° MODIS tile. In the rare case a pixel burned twice during a single fire season (<1%), we retained only the earliest burn date. This approach results in a small reduction of total burned area in order to create standardized annual data layers in both gridded raster and shapefile formats. To locate candidate ignition points within each burned area cluster, we mapped the "local minima," defined as a single grid cell or group of adjacent grid cells with the same
burn date surrounded by grid cells with later burn dates. However, because of variability in orbital coverage and cloud cover, burn date estimates are somewhat uncertain (Giglio et al., 2013), which results in many local minima that may not correspond to actual ignition points. We applied a three-step procedure to address burn date uncertainty and distinguish individual fires. First, we developed a filter to adjust the burn date of local minima that do not correspond to ignition points. Second, we set a "fire persistence" threshold that
determines how long a fire may take to spread from one 500 m grid cell into the next, to distinguish individual fires that are adjacent but occurred at different times in the same fire season. Third, we developed a second filter to correct for outliers in the burn date that occurred along the edges of large fires. Each of these steps is described in detail below.

The ignition point filter is based on the assumption that the fires progress continuously through time and space. First, all local minima were mapped within the original field of burn dates (Fig. 2a and b). Next, each local minimum was replaced by the nearest later burn date in time of the surrounding grid cells, and a new map of local minima was created. If the original local minimum remained as a part of a new, larger local minimum with a later burn date, the fire followed a logical progression in time and space and the
original local minimum was retained. If the local minimum disappeared, the original local minimum was likely the product of an inconsistency within the field of burn dates rather than a true ignition point and the burn date was adjusted forward in time to remove the original local minimum. This step can be repeated several times, with each new iteration further reducing the number of local minima and increasing the confidence in ignition points, yet each iteration also results in a greater adjustment of the original burn date
information (Fig. A1). Here we implemented three iterations of the ignition point filter to remove most local minima that did not spread forward in time while limiting the scope of burn date adjustments (Figs. 2c and d, A1 and A2). For short duration fires, the ignition points were retained associated with the largest possible number of iterations. In all cases, if several local minima were connected through a single cluster of grid cells with the same burn date, only the local minimum with the earliest burn date or largest number
of grid cells was retained, unless the required adjustment of the burn date was larger than the specified burn date uncertainty in the MCD64A1 product. If the final ignition location existed of multiple 500 m grid cells, we used the center coordinates to produce the ignition point shapefile. By design, the ignition point filter cannot adjust the earliest burn date of a fire, and thus has no influence on estimated fire duration.

To establish the location and date of ignition points, as well as to track the daily growth and extent of individual fires, we used a "fire persistence" threshold that determined how long a fire may take to spread

from one 500 m grid cell into the next, taking both fire spread rate and satellite coverage into account (Fig. A3). For example, if an ignition point was adjacent to a fire that burned earlier in the season, this threshold allowed the ignition point to be mapped as separate local minima despite the presence of adjacent burned grid cells with earlier burn dates. On the other hand, if an active fire is covered by dense clouds or smoke, multiple days can pass before a new observation can be made, resulting in a break in fire continuity and increasing the risk of artificially splitting single fires into multiple parts. Using such a threshold is particularly important to distinguish individual fires in frequently burning savannas and highly fragmented agricultural landscapes, where many individual small fires may occur within a relatively short time span. Because there are no reference datasets on global fire persistence, we used a spatially-varying fire persistence threshold that depends on fire frequency (Andela et al., 2017). We assumed that frequently-burning landscapes are generally characterized by faster fires and higher ignition densities, increasing the likelihood of having multiple ignition points within large burn patches, while infrequently burning landscapes will generally be characterized by slower fire spread rates and/or fewer ignitions. In addition, frequently burning landscapes often have a pronounced dry season characterized by low cloud cover, while infrequently burning landscapes may experience a shorter dry season with greater obscuration by clouds. Therefore, we used a 4-day fire persistence threshold for 500 m grid cells that burned more than 3 times during the study period (2003 - 2016), and a 6, 8 and 10-day fire persistence period for grid cells that burned 3 times, 2 times, or 1 time, respectively. These threshold values broadly correspond to biomes, with shorter persistence values for tropical regions and human-dominated landscapes, and longer threshold values for temperate and boreal ecosystems with high fuel loads (Fig. A3).

Based on the location and date of the established ignition points and the fire persistence thresholds, we tracked the growth of each individual fire through time to determine its size, perimeter, and duration (Fig 2f). For each day of year, we allowed individual fires to grow into the areas that burned on that specific day, as long as the difference in burn dates between two pixels was equal to or smaller than the fire persistence threshold of the pixel of origin. When two actively burning fires meet, as on day 255 for the example fires shown in Fig. 2, grid cells that burned on the day of the merger were divided based on nearest distance to the fire perimeter on the previous day.

Burn date uncertainty may also lead to multiple "extinction points," outliers in the estimated day of burn along the edges of a fire. Environmental conditions such as cloud cover complicate the precise estimation of the date of fire extinction, as rainfall events extinguish many fires, and pixels at the edge of the fire may be partially burned and therefore harder to detect. In addition, the contextual relabeling phase of the MCD64A1 algorithm increases burn date uncertainty for extinction points based on a longer consistency threshold (Giglio et al., 2009). We used a second filtering step to adjust the burn date for extinction points, if required. Outliers were adjusted to the nearest burn date back in time, if (1) they represented a cluster no more than 1 to 4 grid cells ($0.21 - 0.9$ km$^2$) along the edge of a fire that was as least 10 times larger and (2) the difference in burn dates was larger than the fire persistence threshold of the adjacent grid cells and thus mapped as a new fire along the edge of the larger fire. If these criteria were met, the outliers were adjusted to the nearest burn date back in time, and incorporated within the larger neighboring fire. However, if these criteria were not met (e.g., for burned areas larger than 4 grid cells), the original burn dates and ignition points were left unadjusted, resulting in separate fires. For the example fires shown in Fig. 2, the adjustment of these outliers affected four grid cells (Fig. 2e) and effectively reduced the number of ignition points (and resulting individual fires) from five (Fig. 2d) to two (Fig. 2f). After adjusting these outliers (extinction points), and including them within the larger fires, we estimated the size (km$^2$), duration (days) and perimeter (km) of each individual fire based on the adjusted burn dates.

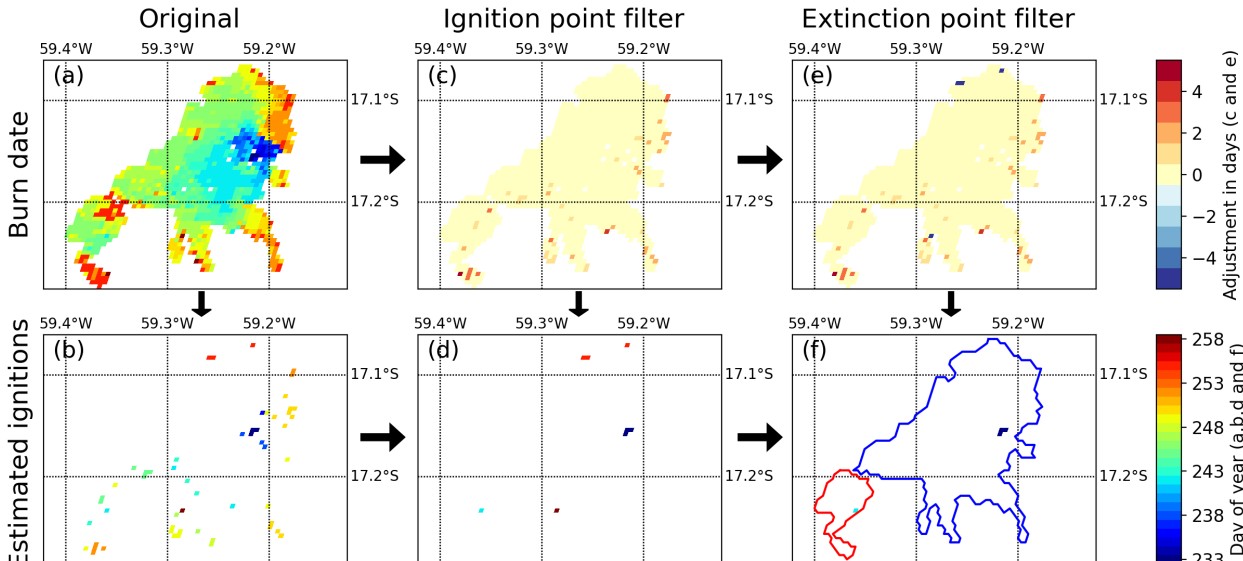

**Figure 2: Algorithm example accounting for uncertainty in the "day of burn" and identifying individual fires within large clusters of adjacent burned pixels.** (a) The original MCD64A1 collection 6 day of burn for one burned patch in the Brazilian Cerrado (in the year 2015), and (b) local minima or "ignition points" identified within the original day of burn data layer. (c) Burn date adjustment based on the filter that removes local minima that do not progress continuously through time and space (positive adjustment), and (d) the corresponding estimate of ignition points based on the adjusted day of burn field. (e) Further burn date adjustment based on the removal of outliers along the edge of the fire (negative adjustment of extinction points), and (f) the final estimate of ignition locations and date by the Global Fire Atlas based on the combined adjustments shown in (e). In (f), the red and blue lines indicate the final fire perimeters.

## 2.2 Daily fire expansion: fire line, speed, and direction of spread

The revised day of burn estimates were used to track the daily expansion ($km^2\ day^{-1}$) and length of the fire line (km) for each individual fire. The daily estimates of fire line length were based on the daily perimeter of the fire, where we assumed that once the fire reached the edge of the burn scar, this part of the perimeter stops burning after one day (Fig. 3a). The expansion of the fire ($km^2\ day^{-1}$) is the area burned by a fire each day. The average speed of the fire line ($km\ day^{-1}$) can now be calculated as the expansion ($km^2\ day^{-1}$) divided by the length of the fire line (km) on the same day. However, this estimate of fire line includes the head, flank and backfire, while it is typically the head-fire that moves fastest and may be responsible for most of the burned area. Moreover, fire dynamics tend to be highly variable in space and time. To understand the spatial variability and distribution of fire speeds, we therefore used an alternative method to estimate the speed and direction of fire spread for each individual 500 m grid cell.

To estimate the speed and direction of spread (Fig. 3), we calculated the most likely path of the fire to reach each individual 500 m grid cell based on shortest distance. More specifically, for each grid cell we estimated the shortest route to connect the grid cell between two points: 1) the nearest point on the fire line with the same day of burn and 2) the nearest point on the previous day's fire line. This route was forced to follow areas burned on the specific day. For each point on this route, or "fire path," the speed of the fire ($km\ day^{-1}$) was estimated as the length of the path (km) divided by one day ($day^{-1}$) and the direction as the direction of the next grid cell on the fire path. Since each grid cell is surrounded by 8 other grid cells, this resulted in eight possible spread directions: north, northeast, east, southeast, south, southwest, west, and northwest. For ignition points that represented a cluster of 500 m grid cells with the same burn date, we assumed that

the fire originated in the center point of the cluster (pixel with largest distance to the final fire perimeter by the end of day 1) and spreads towards the perimeter of the fire by the end of day 1 over the course of one day. For single pixel fires, we assumed the fire burned across 463 m (1 pixel) during a single day and we did not assign a direction of spread. Similarly, fires of all sizes that burned on a single day were not assigned a direction of spread. We corrected estimates of both speed and direction for the orientation between 500 m grid cells on the MODIS sinusoidal projection that varies with location. When a particular grid cell formed part of multiple "fire paths," the earliest time of arrival or the highest fire speed and corresponding direction of spread were retained. This assures a logical progression of the fire in time and space and corresponds to fires typically moving fastest in a principal direction and then spreading more slowly along the flank.

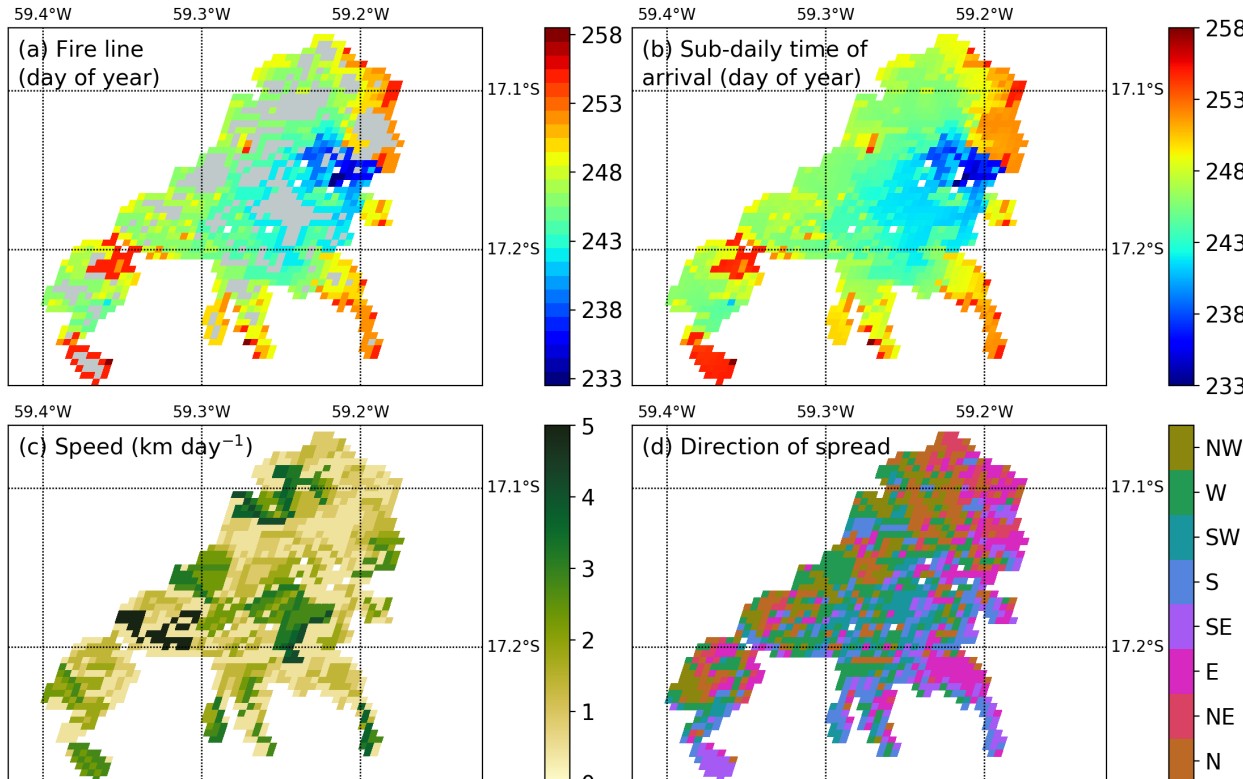

**Figure 3: Sub-daily estimates of fire progression can be used to estimate spatiotemporal variation in fire speed and direction of spread.** (a) daily progression of the fire line, (b) interpolated estimates of sub-daily time of arrival, (c) fire speed (km day$^{-1}$), and (d) direction of spread. The light gray areas in (a) are burned areas between fire lines and correspond to areas of relatively high fire speed. White areas were not burned.

## 2.3 Preliminary accuracy assessment

Few large-scale datasets are available on daily or sub-daily fire dynamics, highlighting the novelty of the Global Fire Atlas dataset but also posing challenges for validation. Here we used four alternative datasets to carry out an initial accuracy assessment. First, we used active fire detections to assess the temporal accuracy of the Global Fire Atlas burn date. Second, we compared fire perimeters to independent fire perimeter data for the continental US. Third, we combined the independent data on fire perimeters with active fire detections to evaluate the Global Fire Atlas fire duration estimates. Finally, we compared Global Fire Atlas data to a small (manually compiled) dataset of daily fire perimeters from the US Forest Service.

To evaluate burn dates in the Global Fire Atlas, we used the 375 m resolution active fire detections (VNP14IMGML C1) derived from the Visible Infrared Imaging Radiometer Suite (VIIRS) instrument aboard the Suomi National Polar-orbiting Partnership (Suomi-NPP) satellite (Schroeder et al., 2014). Active fire detections provide accurate information on the burn date, particularly in ecosystems with low fuel loads where fires will typically be active during only a single day in each particular grid cell. We compared the date of active fire detections from VIIRS within each larger 500 m MODIS grid cell (based on VIIRS center point) to the adjusted MCD64A1 day of burn to understand the temporal precision of the derived Global Fire Atlas products. If several active fire detections were available for a single 500 m MODIS grid cell, we reported the day closest to the temporal mean. We compared all 500 m MODIS grid cells with corresponding active fire detection during the overlapping data period (2012 – 2016) for four different ecosystems globally: (1) forests (including all forests), (2) shrublands (including open and closed shrublands), (3) woody savannas, and (4) savannas and grasslands, with land cover type derived from MODIS MCD12Q1 collection 5.1 data for 2012 using the University of Maryland (UMD) classification (Friedl et al., 2002).

We compared fire perimeters from the Global Fire Atlas to fire perimeter estimates from the Monitoring Trends in Burn Severity (MTBS) project during their overlapping period (2003 – 2015). The MTBS project provides semi-automated estimates of fire perimeters based on 30 m Landsat data for fires with a minimum size of 1000 acres (405 ha) in the western US and 500 acres (202 ha) in the eastern US (Eidenshink et al., 2007; Sparks et al., 2015). To determine overlap between MTBS and Fire Atlas perimeter estimates, we rasterized the MTBS perimeters onto the 500 m MODIS sinusoidal grid, including all 500 m grid cells with their center point within the higher resolution (30 m) MTBS fire perimeter. For all overlapping fire perimeters, we compared the original MTBS fire perimeter information with the Fire Atlas estimates of fire perimeters. In cases with multiple overlapping perimeters, fires with the largest overlapping surface area were compared.

We also combined MTBS fire perimeters with VIIRS active fire detections to derive an alternative estimate of fire duration (2012 – 2015). To estimate fire duration from these products, we first determined the median burn date of each fire according to the MCD64A1 burned area data. Subsequently, we included all VIIRS active fire detections before and after the median or 'center' burn date until a period of three fire-free days was reached. Any active fire detections that occurred outside this timeframe were excluded to avoid overestimation of the fire duration due to smoldering or possible false detections before or after the fire. Two thresholds were used to select a subset of MTBS and Fire Atlas perimeters to assess the accuracy of estimated fire duration. Fires were first matched based on perimeters, with a maximum tolerance of a threefold difference in length between perimeters. Second, we further selected MTBS perimeters with VIIRS active fire detections for at least 25% of the 500 m Fire Atlas grid cells. These thresholds excluded 51% of the overlapping fire perimeters, but reduced errors originating from cloud cover or differences in the underlying burned area estimates (e.g., resolution, methodology) to evaluate estimated fire duration. Similar to the assessment of burn date accuracy, comparisons of fire perimeters and fire duration with MTBS data over the continental US were grouped into four land cover types: (1) forests, (2) shrublands, (3) woody savannas, and (4) savannas and grasslands.

For specific large wildfires across the western USA, the US Forest Service National Infrared Operations (NIROPS; https://fsapps.nwcg.gov/nirops/) estimates daily fire perimeters for fire management purposes by collecting aircraft high resolution infrared imagery. This imagery is manually analyzed by trained specialists to extract the active fire front. Although these data provide a wealth of information, only a small number of fires are completely and precisely documented. We were able to extract 15 large fires from the NIROPS database for which daily perimeter information was available. Although insufficient for full scale validation, the comparison with NIROPS data provides valuable insights into the strengths and shortcomings of the Global Fire Atlas estimates of individual fire size, duration and expansion rates. In addition to per fire averages, we compared day-to-day expansion rates ($km^2$ $day^{-1}$) of individual large fires

across both datasets. If multiple Global Fire Atlas perimeters overlapped with a single US Forest Service fire perimeter, we compared the fires with the largest overlapping surface area.

## 3 Results

### 3.1 Preliminary accuracy assessment

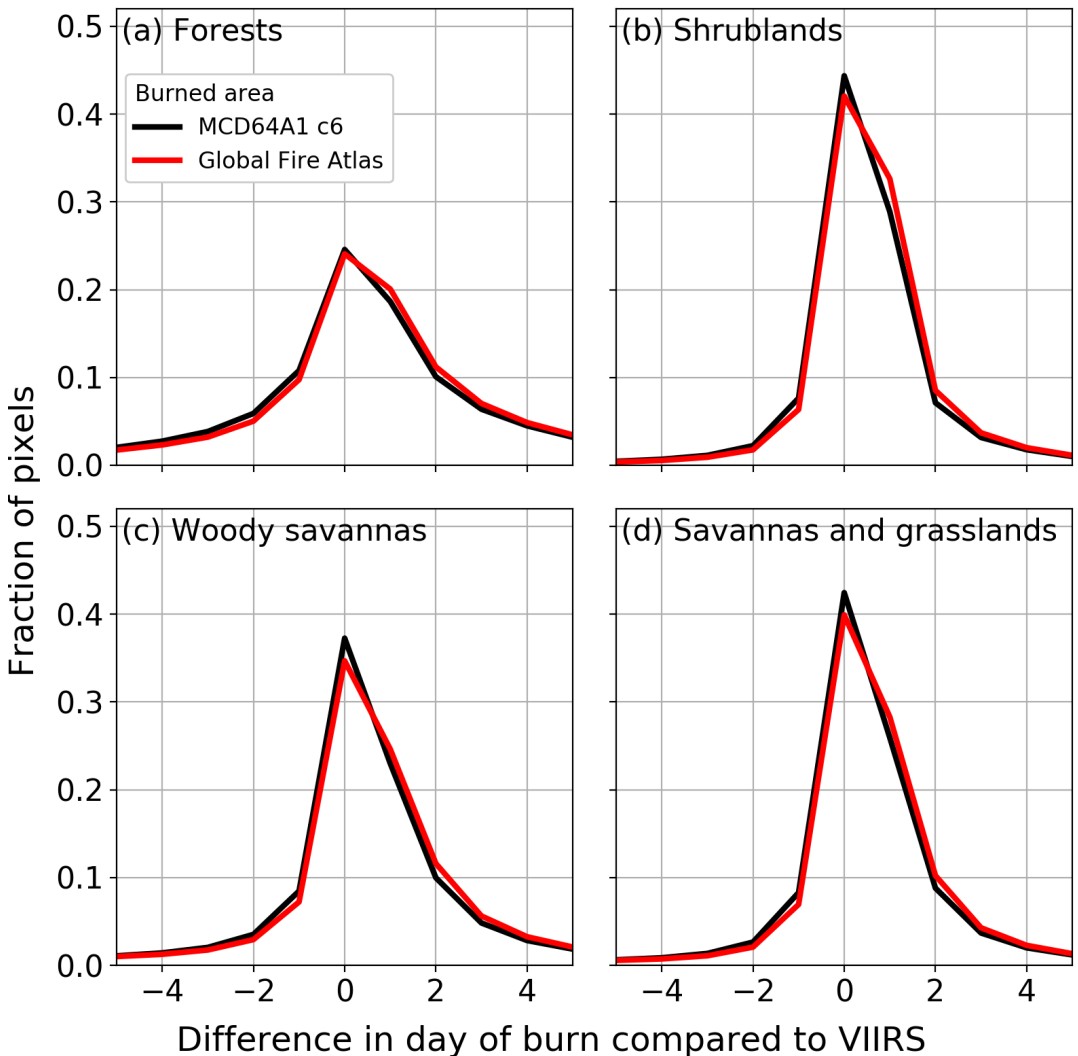

**Figure 4: Global comparison of burn dates derived from the MCD64A1 burned area product, adjusted burn dates of the Global Fire Atlas, and VIIRS active fire detections (2012 − 2016).** (a) Forests, (b) shrublands, (c) woody savannas, and (d) savannas and grasslands. Negative values indicate pixels with a burned area day of burn earlier than the corresponding VIIRS active fire detection, zero indicates no difference in day of burn between both datasets, and positive numbers indicate a delayed detection of burned area compared to active fire detections.

At the pixel scale, estimated burn dates from burned area and active fire products were comparable (Fig. 4), with greater variability across biomes than from minor burn date adjustments in the Global Fire Atlas algorithm. Burn dates estimated from MODIS burned area and VIIRS active fire detections were least

comparable in high-biomass ecosystems with lower fire spread rates. In forests and woody savannas 24% and 35% of burned pixels were detected on the same day and 54% and 67% within ± 1 day, respectively (Fig. 4a and c). With decreasing biomass, the direct correspondence between burn dates from burned area and active fire detections increased to 41% (same day) and 80% (± 1 day) in shrublands (Fig. 4b) and 40% (same day) and 75% (± 1 day) in savannas and grasslands (Fig. 4d). These differences likely stem from the

combined increase in uncertainty of burn date in higher-biomass ecosystems and influence of fire persistence (multiple active fire days in a single 500 m grid cell) on the ability to reconcile the timing of burned area and active fire detections in these ecosystems. Several factors may account for the positive bias in the 500 m day of burn from burned area compared to active fire detections, including orbital coverage, cloud and smoke obscuration, and different thresholds between burned area and active fire algorithms

regarding the burned fraction of a 500 m grid cell. The adjustments we made to the burn date in the Global Fire Atlas required to effectively determine the extent and duration of individual fires, had a relatively small effect on the overall accuracy assessment but tended to reduce the negative bias in burn dates and increase the positive bias compared to the underlying MCD64A1 c6 product (see red and black lines in Fig 4). In line with these findings, we found good agreement between a 3-day running average of Global Fire Atlas

and US Forest service estimates of daily fire expansion, but reduced correspondence for daily estimates of fire growth rates due to uncertainty in the day-of-burn of the burned area product (Fig. B1).

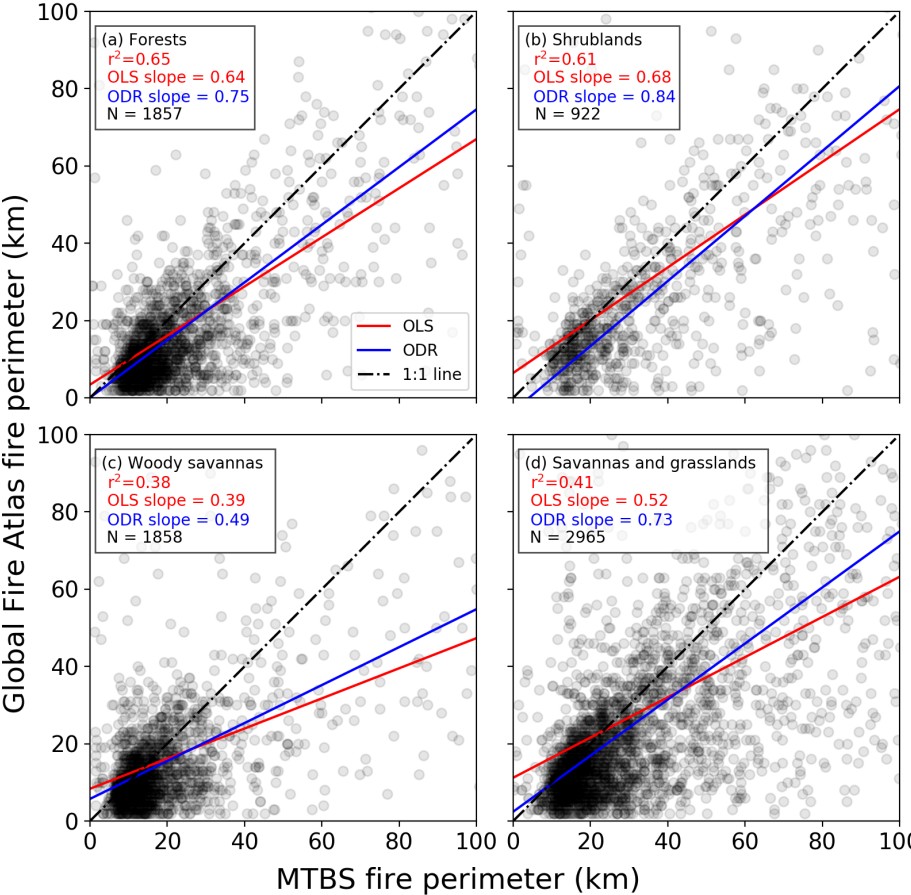

**Figure 5: Comparison of fire perimeter estimates based on the Global Fire Atlas and MTBS for the**
**continental US (2003 – 2015).** (a) Forests, (b) shrublands, (c) woody savannas, and (d) savannas and grasslands. Red lines indicate the slope between both datasets based on ordinary least squares (OLS) with corresponding $r^2$ values, while blue lines are based on orthogonal distance regression (ODR). For the scatter plots, darker gray or black indicates a greater density of points.

For fire perimeters, the best agreement between the Global Fire Atlas and MTBS was found in forests and shrublands, where the Global Fire Atlas reproduced 65% and 61% of the observed variance in MTBS fire perimeters, respectively (Fig. 5). Less agreement was found for woody savannas (38%) and savannas and grasslands (41%). Overall, the Global Fire Atlas underestimated fire perimeter length in all of the vegetation classes. However, uncertainty exists in both datasets. Orthogonal distance regression (ODR) accommodates

uncertainties in both datasets and generally resulted in slopes closer to the 1:1 line, indicating closer correspondence, on average, in absolute perimeter estimates for the two datasets. An in-depth comparison of the performance of the Global Fire Atlas and the MTBS datasets for several grassland fires in Kansas (USA) suggested that differences originated both from the underlying burned area datasets and the methodologies (Fig. B2). For this particular grassland in Kansas, the MCD64A1 product estimated less

burned area compared to the Landsat-based MTBS dataset, resulting in fragmentation of larger burn scars into disconnected patches. However, the daily temporal resolution of the MCD64A1 burned area product allowed for recognition of individual ignition points within larger burn patches of fast-moving grassland fires that cannot be separated using infrequent Landsat imagery (Fig. B2). In addition, the 30 m spatial resolution of the MTBS perimeters may result in more irregularity and therefore in longer fire perimeter

estimates compared to the 500 m resolution Fire Atlas perimeters. Combined, these tradeoffs in spatial and temporal resolution resulted in less agreement between fire perimeters in woody savannas (Fig. 5c) and savannas and grasslands (Fig. 5d).

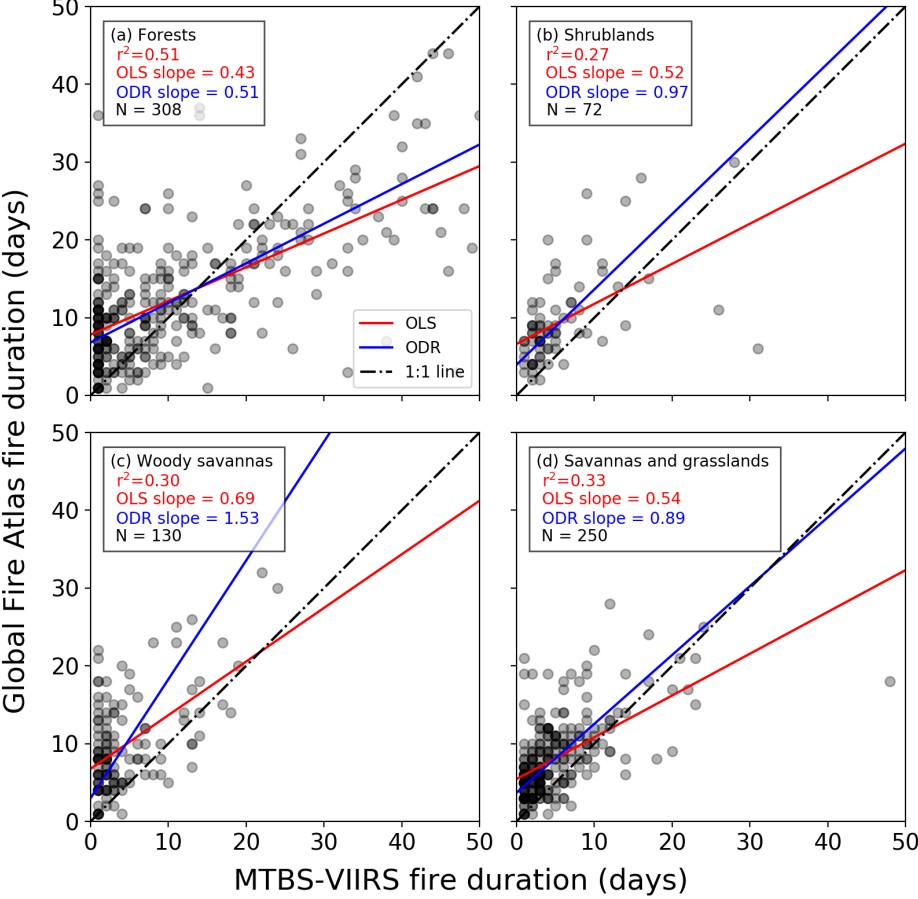

**Figure 6: Comparison of fire duration estimates from the Global Fire Atlas and the combination of VIIRS active fire detections within MTBS fire perimeters for the continental US (2012 – 2015).** (a) Forests, (b) shrublands, (c) woody savannas, and (d) savannas and grasslands. Red lines indicate the slope

between both datasets based on ordinary least squares (OLS) with corresponding $r^2$ values, while blue lines are based on orthogonal distance regression (ODR). For the scatter plots, darker gray or black indicates a greater density of points. This comparison was made for a subset of MTBS and Global Fire Atlas perimeters using selection criteria for perimeter overlap and VIIRS active fire detections described in Section 2.3.

Initial assessment of the accuracy of fire duration estimates from the Global Fire Atlas highlighted differences in the sensitivity of satellite-based burned area and active fire products to fire lifetime (Fig. 6). Similar to fire perimeters, the best agreement in fire duration estimates was found for forests, where the Global Fire Atlas reproduced 51% of the observed variance of the fire duration estimates based on combining MTBS fire perimeters with active fire detections. Shrublands, woody savannas, and savannas and grasslands had lower correlations, with 27%, 30% and 33% of the variance explained, respectively. The orthogonal distance regression resulted in slopes close to the one-to-one line for shrublands and savannas and grasslands, indicating reasonable agreement. Fire duration was clearly underestimated for forested ecosystems with high fuel loads, as fires may continue to smolder for days (resulting in active fire detections) after the fire has stopped expanding.

The comparison of Global Fire Atlas data to a small dataset (n = 15) of daily perimeters of large wildfires in primarily forested cover types mapped by the US Forest Service yielded good correspondence between estimates of fire size, duration, and expansion rates (Fig. 7). The improved comparison of fire size (cf. Fig. 5a and 7a) could be related to the US Forest Service data being more accurate than MTBS, but likely also represents the good performance of the Global Fire Atlas (e.g. compare Figs. 7a, b and c to Figs. 7d, e and f) and underlying burned area products (Fusco et al., 2019) for relatively large fires. In contrast to the suggested underestimate of fire duration shown in Fig. 6a, these data suggest the Global Fire Atlas may slightly overestimate fire duration. This difference may reflect the fact that active fire detections may be triggered by smoldering while the burned area product will only register the initial changes in surface reflectance from fire. Both comparisons (Figs. 6, 7b and 7e) suggest the Global Fire Atlas may overestimate the duration of smaller fires with relatively short duration, likely based on the uncertainty in underlying burn dates. Based on a small underestimate of overall burned area and overestimate of fire duration by the Global Fire Atlas, the average daily fire expansion rates based on US Forest Service data were higher than estimates based on Global Fire Atlas data (Fig. 7c and f).

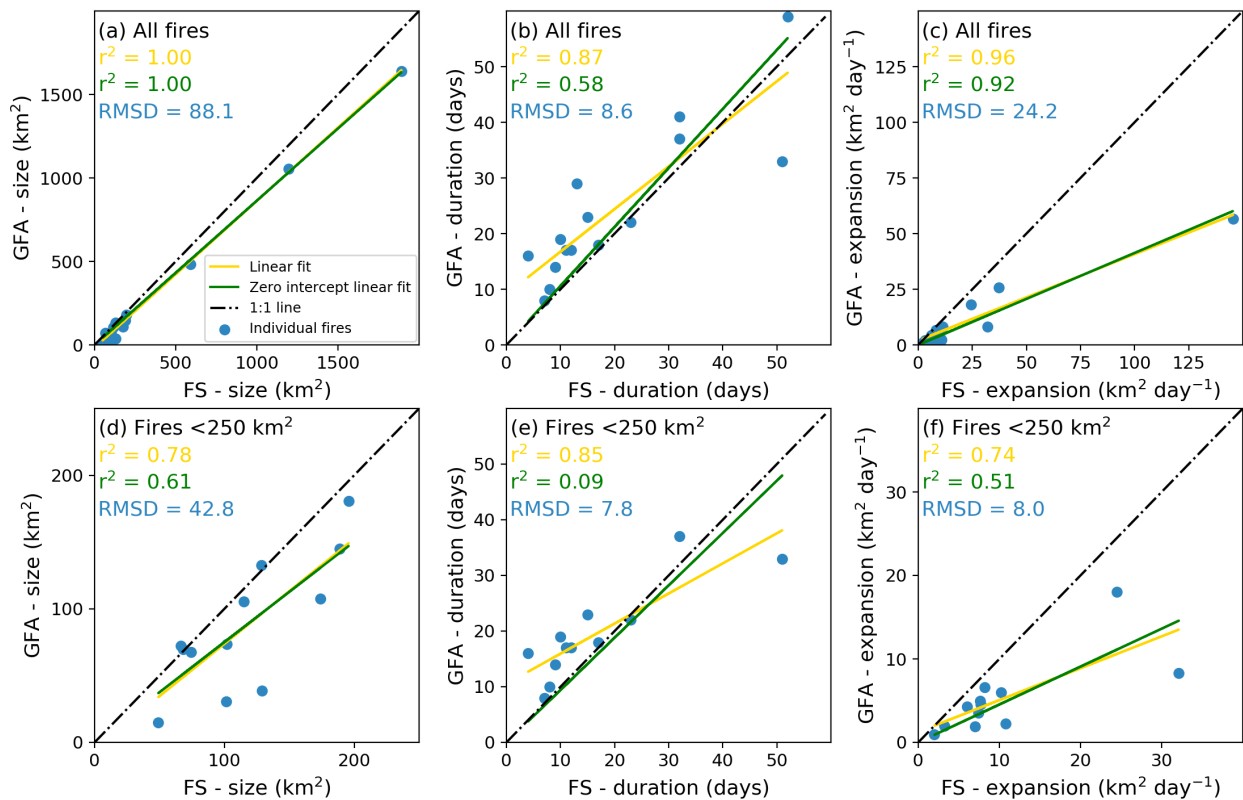

**Figure 7: Comparison of Global Fire Atlas (GFA) and US Forest Service (FS) data for a selected number of large wildfires in the US.** Comparison of (a) fire size, (b) duration, and (c) average daily expansion rate for all fires (N=15), (d, e and f) are like (a, b and c) but for fires smaller than 250 km$^2$ (N=12). Correlation coefficients are provided based on linear regression with (yellow) and without (green) an intercept, assuming a non-zero intercept could indicate a structural offset between both datasets. Root-mean-square deviations (RMSD) are reported in blue.

## 3.2 Characterizing global fire regimes

Over the 14-year study period we identified 13,250,145 individual fires with an average size of 4.4 km$^2$ (Table 1) and minimum size of one MODIS pixel (21 ha or 0.21 km$^2$). On average, largest fires were found in Australia (17.9 km$^2$), boreal North America (6.0 km$^2$), and northern hemisphere Africa (5.1 km$^2$), while central America (1.7 km$^2$), equatorial Asia (1.8 km$^2$), and Europe (2.0 km$^2$) had the smallest average fire sizes (Table 1). Spatial patterns of the number of ignitions and fire sizes were markedly different and often inversely related (Fig. 8). Burned area in agricultural regions and parts of the humid tropics, particularly in Africa, resulted from high densities of fire ignitions and relatively small fires, consistent with widespread use of fire for land management. Large fires accounted for most of the burned area in arid regions, high latitudes, and other natural areas with low population densities and a sufficiently long season of favorable fire weather (Fig. 8).

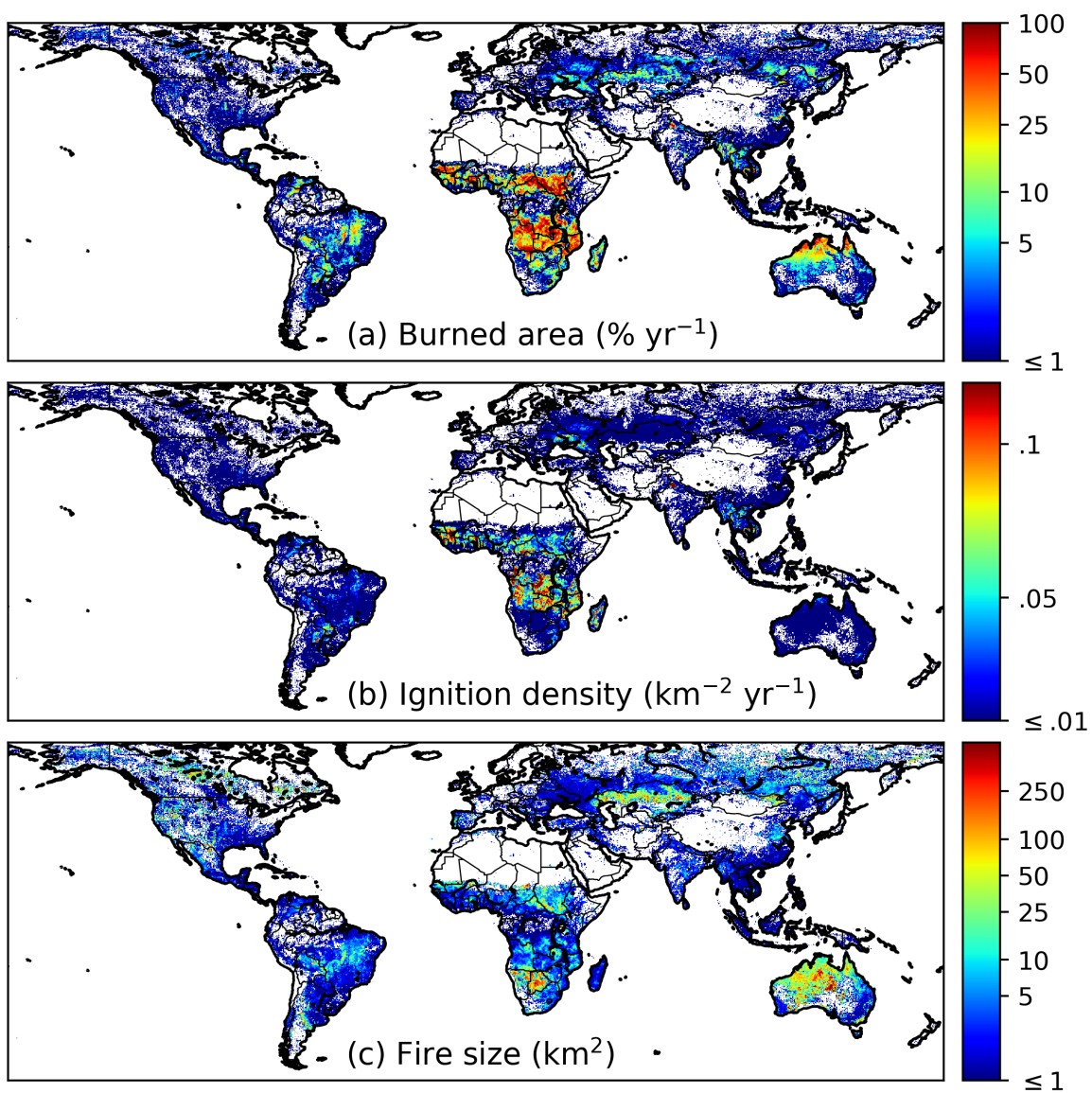


**Figure 8: Average global burned area (MCD64A1), ignition density, and fire size over the study period 2003 – 2016.** For any given location, burned area in panel (a) can be represented as the product of ignitions per year shown in (b) and fire size shown in (c).

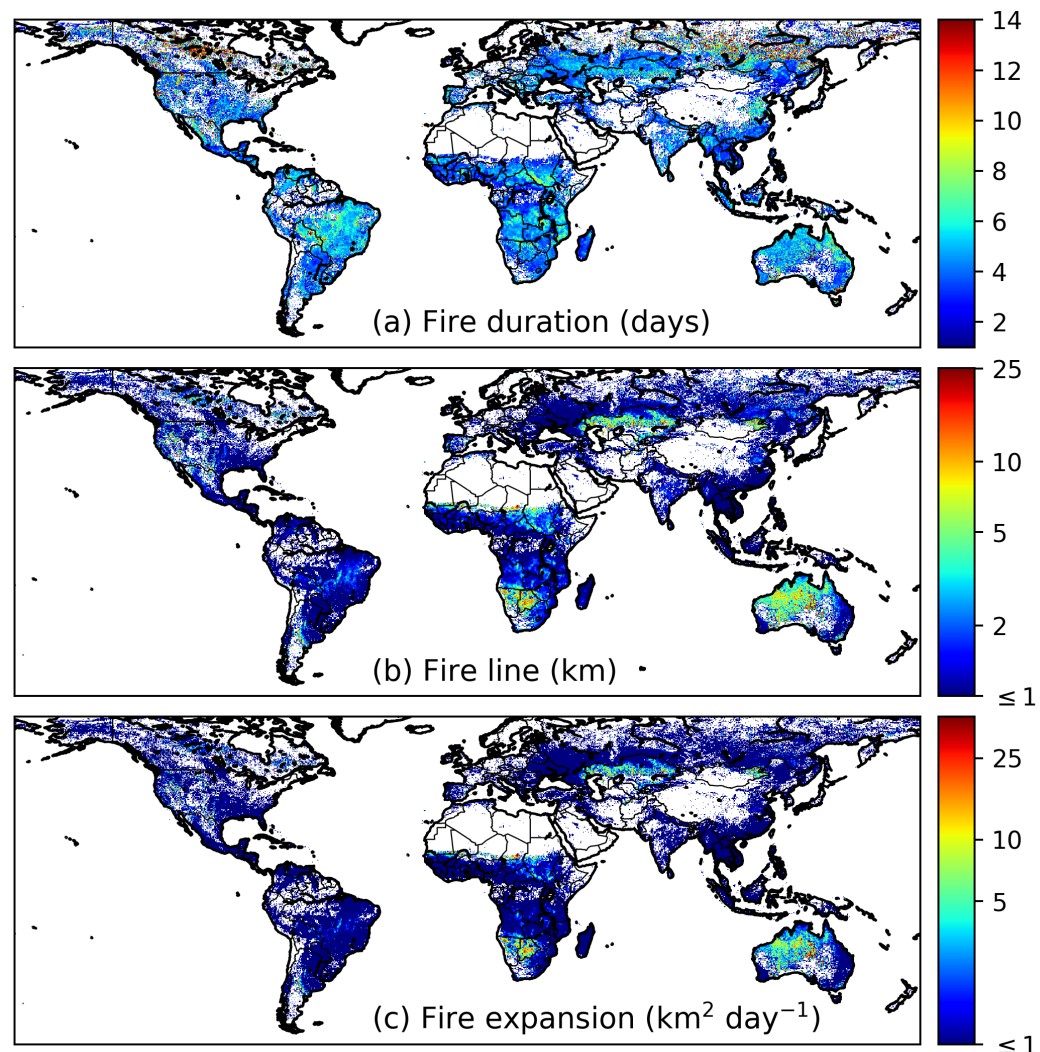


**Figure 9: Average fire duration (a), fire line length (b), and daily expansion (c) over the study period 2003 – 2016.** Fire size (see Fig. 7c) is the product of fire duration (a) and daily fire expansion (c).

Global patterns of fire duration and expansion rates provide new insight about the occurrence of large fires,
as the size of each fire ($km^2$) is the product of fire duration (days) and daily fire expansion rate ($km^2 day^{-1}$). Individual fires that burned for a week or more occurred frequently across the productive tropical grasslands and in boreal regions (Fig. 9a, Table 2). In these regions, fire duration exerted a strong control on fire size and total burned area. On average, human-dominated landscapes such as deforestation frontiers or agricultural regions experienced smaller and shorter fires compared to natural landscapes (Table 2). Fire
duration was also relatively short in semiarid grasslands and shrublands characterized by high daily fire expansion rates, based on the development of long fire lines (Fig. 9b and c) and high velocity. In these semiarid regions, fire duration and size were likely limited by fuel availability and connectivity. In line with these findings, largest average daily expansion rates were found in Australia (1.7 $km^2 day^{-1}$), northern hemisphere Africa (0.9 $km^2 day^{-1}$) and southern hemisphere Africa (0.9 $km^2 day^{-1}$), and smallest expansion
rates in central America (0.3 $km^2 day^{-1}$), equatorial Asia (0.3 $km^2 day^{-1}$), and southeast Asia (0.4 $km^2 day^{-1}$; Table 1).

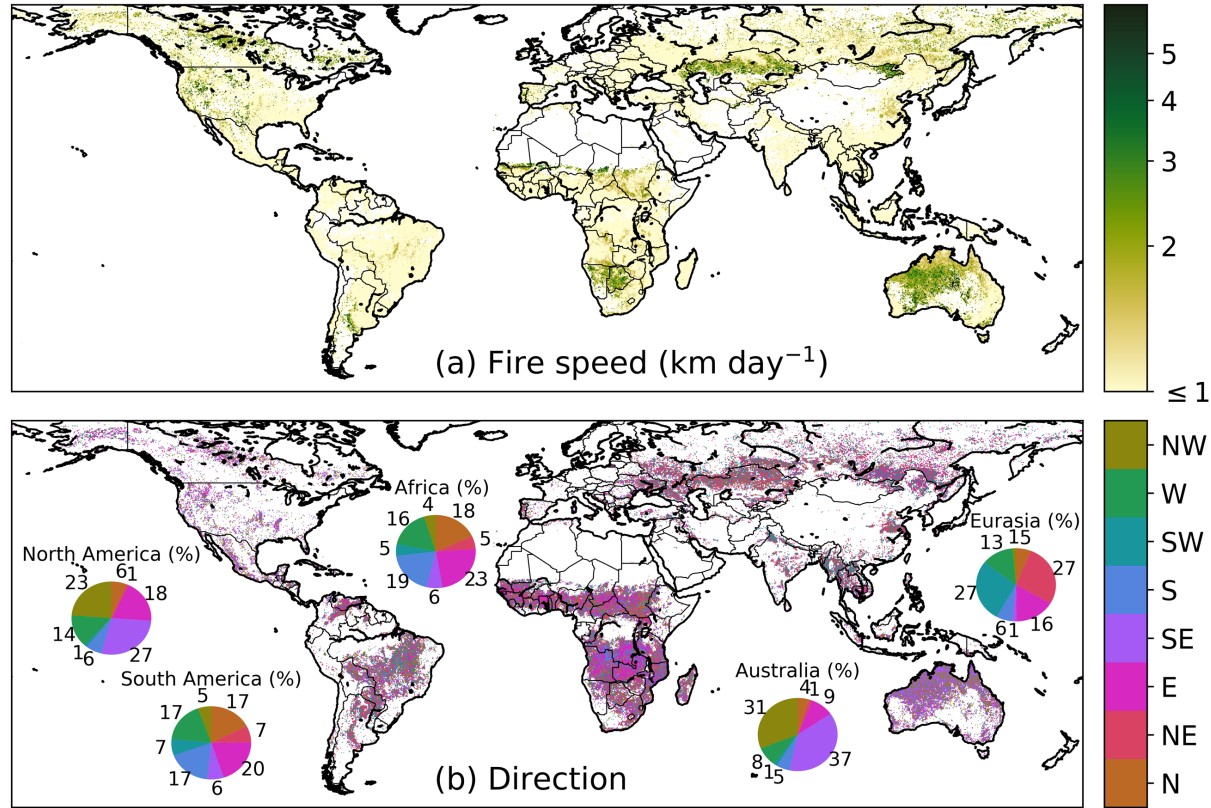

**Figure 10: Average fire speed (a) and the dominant direction of fire spread (b) over the study period 2003 – 2016.** For each 0.25° grid cell the direction was estimated as the dominant fire spread direction of fires larger than 10 km² within the grid cell. We focused on larger fires ($\geq 10$ km²) to determine the dominant spread direction, because large fires will generally express a clearer spatiotemporal structure of fire spread at 500 m daily resolution. Pie charts show the fraction of individual larger fires ($\geq 10$ km²) by dominant spread direction for each continent.

The fastest fires occurred in arid grasslands and shrublands (Fig. 10a), where fuel structure, climate conditions, and emergent properties of large wildfires contribute to high fire spread rates. Relatively high fire speeds were also observed in some parts of the boreal zone, particularly in central and western Canada. Lowest fire velocities were observed in infrequently burning humid tropical regions where fire spread was influenced by higher fuel loads and humidity (Table 1). At all scales, estimated fire direction exhibited considerable complexity (Fig. 10b). With some regional exceptions, no clear dominant spread direction was found in South America or Africa. Based on the underlying 500 m data layers, landscape structure and drainage patterns played an important role in controlling individual fire spread direction in the humid tropics. Fire spread direction also varied considerably within individual fires, and the dominant direction typically represented less than half of the pixels. Fire spread direction was more consistent in the arid tropics, as demonstrated by the northwest and southeast orientation of fire spread in Australia, consistent with the dominant wind directions. At mid-latitudes, we found evidence for more east and westward fire progression in Europe and Asia and northwest and southeast spread direction in North America, broadly consistent with the orientation of mountain ranges and other topographic features within the key biomass burning regions.

**Table 1: Fire attributes for each Global Fire Emissions Database (GFED) region during 2003 – 2016.** Ignitions are the summed ignitions over the study period (2003 – 2016). For size, duration, expansion, and speed we report the mean values for individual fires and also the mean weighted by fire size (the latter estimate is provided in parentheses). For ignitions, regions with over one million ignitions are shown in red and lower values in blue, for other fire aspects values equal to or above the global average are shown in red and below the global average in blue. A map of the GFED regions is shown in the annex material (Fig. B3a).

| GFED Region | Ignitions (2003-2016) | Size (km$^2$) | Duration (days) | Expansion (km$^2$ day$^{-1}$) | Speed (km day$^{-1}$) |
|---|---|---|---|---|---|
| World | 13250145 | 4.4 (395.9) | 4.5 (14.7) | 0.6 (14.5) | 0.9 (3.2) |
| BONA | 57613 | 6.0 (202.8) | 5.4 (23.3) | 0.5 (6.8) | 1.0 (4.3) |
| TENA | 137900 | 2.9 (136.7) | 4.7 (13.4) | 0.5 (8.8) | 0.8 (3.7) |
| CEAM | 229245 | 1.7 (28.3) | 4.3 (12.2) | 0.3 (1.5) | 0.7 (1.4) |
| NHSA | 242359 | 3.1 (50.1) | 5.1 (12.4) | 0.5 (3.3) | 0.8 (2.1) |
| SHSA | 1320177 | 3.0 (90.6) | 4.7 (13.8) | 0.5 (4.8) | 0.7 (2.3) |
| EURO | 71233 | 2.0 (30.7) | 4.6 (10.3) | 0.4 (2.7) | 0.7 (2.0) |
| MIDE | 86783 | 2.3 (22.0) | 4.0 (9.8) | 0.5 (2.1) | 0.8 (1.9) |
| NHAF | 3517808 | 5.1 (186.2) | 4.4 (14.7) | 0.7 (8.6) | 0.9 (3.0) |
| SHAF | 5000436 | 4.3 (232.5) | 4.5 (13.5) | 0.7 (9.6) | 0.9 (2.6) |
| BOAS | 363279 | 3.7 (116.8) | 4.5 (15.6) | 0.5 (6.8) | 1.0 (4.1) |
| CEAS | 807739 | 3.2 (339.7) | 4.2 (11.5) | 0.5 (22.7) | 0.8 (5.6) |
| SEAS | 937810 | 2.2 (27.8) | 4.1 (13.2) | 0.4 (1.8) | 0.7 (1.8) |
| EQAS | 117870 | 1.8 (13.5) | 5.5 (16.4) | 0.3 (0.8) | 0.7 (1.3) |
| AUST | 358807 | 17.9 (2030.6) | 5.0 (20.5) | 1.7 (59.5) | 1.2 (6.1) |

**Table 2: Fire attributes by GFED fire type during 2003 – 2016.** Ignitions are the summed ignitions over the study period (2003 – 2016). For size, duration, expansion, and speed we report the mean values for individual fires and also the mean weighted by fire size (the latter estimate is provided in parentheses). For agriculture, we only included fires with greater than 90% of burned area classified as cropland. For ignitions, fire types with over one million ignitions are shown in red and lower values in blue, for other fire aspects values equal to or above the global average are shown in red and below the global average in blue. A map of the GFED fire types is shown in the annex material (Fig. B3b).

| GFED fire type | Ignitions (2003-2016) | Size (km$^2$) | Duration (days) | Expansion (km$^2$ day$^{-1}$) | Speed (km day$^{-1}$) |
|---|---|---|---|---|---|
| All | 13250145 | 4.4 (395.9) | 4.5 (14.7) | 0.6 (14.5) | 0.9 (3.2) |
| Boreal forest | 197124 | 5.2 (149.2) | 5.4 (20.1) | 0.6 (6.5) | 1.0 (4.2) |
| Temporal forest | 178909 | 2.5 (84.1) | 4.1 (14.0) | 0.4 (4.2) | 0.8 (2.8) |
| Deforestation | 909826 | 1.4 (28.7) | 3.8 (13.7) | 0.3 (1.4) | 0.6 (1.4) |
| Savanna | 9809719 | 5.1 (447.5) | 4.6 (14.9) | 0.7 (16.2) | 0.9 (3.4) |
| Agriculture | 1631918 | 1.4 (26.4) | 3.4 (10.3) | 0.3 (2.0) | 0.7 (1.9) |

## 3.3 Fire extremes

The world's largest individual fires were mostly found in sparsely populated arid and semiarid grasslands and shrublands of interior Australia, Africa, and Central Asia (Fig. 11a). Strikingly, fires of these proportions were nearly absent in North and South America, possibly due to higher landscape fragmentation

and different management practices, including active fire suppression. In arid regions of Southern Africa and Australia, large fires typically followed La Niña periods (e.g., 2011 and 2012), when increased rainfall and productivity increase fuel connectivity (Chen et al., 2017). The largest fire in the Global Fire Atlas occurred in northern Australia, burning across 40,026 km$^2$ (about the size of Switzerland or the Netherlands) over a period of 72 days with an average speed of 19 km day$^{-1}$, following the 2007 La Niña. The longest fires burned for over 2 months in seasonal regions of the humid tropics and high-latitude forests (Fig. 11b). Drought conditions in 2007 and 2010 caused multiple fires to burn synchronously for over two months across tropical forests and savannas in South America. The highest fire velocities typically occurred in areas of low fuel loads. While fires larger than 2500 km$^2$ were nearly absent from arid grass and shrublands in North and South America, patterns of extremely fast-moving fires in arid grass and shrublands were similar to other continents. Fast-moving fires also show evidence of synchronization, for example with several extremely fast fires burning across the steppe of eastern Kazakhstan during 2003 (Fig. 11c).

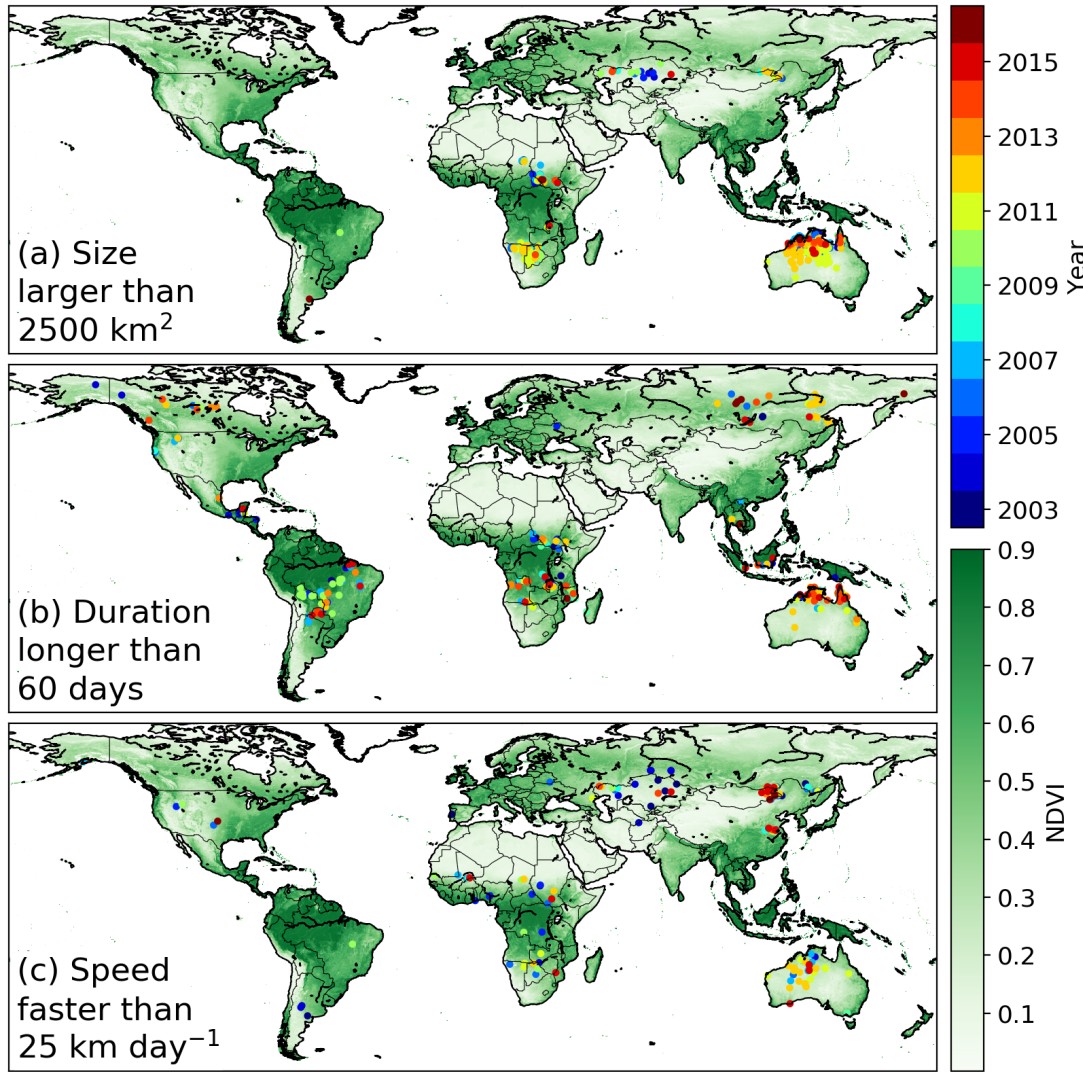

**Figure 11: Location and year of the largest, longest, and fastest fires over the study period 2003 – 2016.** (a) fires larger than 2500 km$^2$, (b) fires longer than 60 days, and (c) fires with an average velocity higher than 25 km day$^{-1}$. The background image depicts mean MODIS normalized difference vegetation index (NDVI, 2003 – 2016), an indicator for large scale vegetation patterns and available fuels.

# 4 Discussion

The Global Fire Atlas is the first freely available global dataset to provide daily information on seven key fire characteristics: ignition timing and location, fire size, duration, daily expansion, daily fire line, speed and direction of spread based on moderate resolution burned area data. Over the 2003 – 2016 study period, we identified over 13 million individual fires ($\geq 21$ ha) (Table 1). Characteristics of these fires varied widely across ecosystems and land use types. In arid regions and other fire-prone natural landscapes, most of the burned area resulted from a small number of large fires (Fig. 8). Fire sizes declined along gradients of increasing rainfall and human activity, with larger numbers of small fires in the humid tropics or other human-dominated landscapes. Multiday fires were the norm across nearly all landscapes, with some large fires in productive tropical grasslands and boreal regions burning for over two months during drought periods (Fig. 11). The dominant control on fire size also varied across ecosystems; fire duration was the principal control on fire size in boreal forests, whereas fuels limited the size of fast-moving fires in arid grasslands and shrublands (Figs. 9 and 10). Characterizing fire behavior across large scales is key for understanding fire-vegetation feedbacks, emissions estimates, fire prediction, effective fire management, and building mechanistic models of fires within ecosystem models. Satellite remote sensing has been widely used to characterize global pyrogeography (Archibald et al., 2013) and fire-climate interactions (Westerling et al., 2006; Alencar et al., 2011; Morton et al., 2013a; Field et al., 2016; Young et al., 2017). Despite this progress, large-scale understanding of individual fire behavior has remained limited by the availability of consistent global-scale data products. Analysis and future refinement of the Global Fire Atlas may be useful in this context, providing new insight about the response of fires to different global change drivers.

Both climate and human activity exert a strong control on global burned area (Bowman et al., 2009) and contribute to rapidly changing fire regimes worldwide (Jolly et al., 2015; Andela et al., 2017; Earl and Simmonds, 2018). Moreover, increasing human presence in fire prone ecosystems requires increased efforts to actively manage fires for ecosystem conservation and human wellbeing (Moritz et al., 2014; Knorr et al., 2016). The ignition location, spread, and duration of individual fires can be used to address new questions in the field of fire-climate interactions and the changing influence of human activity on fire behavior, as each of these metrics may respond differently to variability or change. For example, recent studies have suggested that climate warming and drying may increase fire size and burned area in the tropics (Hantson et al., 2017) and at higher latitudes (Yang et al., 2015). Our findings suggest that an increase in the length of the fire season may be the dominant driver for increases in fire activity in these ecosystems, as fire duration was a strong control on eventual fire sizes and burned area (Figs. 8, 9 and 11). Investigating fire-climate interactions and human controls on burned area using the Fire Atlas data layers will benefit management efforts and science investigations, as fire alters vegetation structure (Bond et al., 2005; Staver et al., 2011), biogeochemical cycles (Bauters et al., 2018; Pellegrini et al., 2018) and climate (Randerson et al., 2006; Ward et al., 2012).

The Global Fire Atlas provides several new constraints that could improve the representation of fires in ecosystem and Earth system models. Fire models embedded in dynamic vegetation models are important tools for understanding the changing role of fires in the Earth system and the ecosystem impacts of fires (Hantson et al., 2016; Rabin et al., 2017). Most global models of fire activity are calibrated using satellite-derived estimates of total burned area or active fires (Hantson et al., 2016), rather than individual fire characteristics such as fire size. As a result, many of these fire models capture the spatial distribution of global fire activity but not burned area trends (Andela et al., 2017) or interannual variability that may occur as a consequence of changes in fire spread rate or duration. Models range from simple empirical schemes to complex, process-based representations of individual fires (Hantson et al., 2016; Rabin et al., 2017). Process-based models estimate burned area as the product of fire ignitions and size, while many models include a dynamic rate of spread to determine eventual fire sizes (e.g. SPITFIRE; Thonicke et al., 2010) but use arbitrary threshold values for key parameters such as fire duration (Hantson et al., 2016). We found

that global patterns of fire duration, ignition, size, and rate of spread (i.e. speed) varied widely across ecosystems and human land management types, and thus these Global Fire Atlas data products provide additional pathways to benchmark models of various levels of complexity. While only a few models include
multiday fires (e.g., Pfeiffer et al., 2013; Le Page et al., 2015; Ward et al., 2018), we found that multiday fires were the norm across most biomes, and fire duration forms an important control on eventual fire sizes and burned area in many natural ecosystems with abundant fuels. Similarly, many models assume relatively homogeneous fuel beds, while our results suggest that landscape features and vegetation patterns result in highly heterogeneous fuel beds that form a strong control on fire spread (speed and direction). Large
differences in fire behavior across ecosystems and management strategies may improve fire emissions estimates and emissions forecasting, particularly when combined with active fire detections to better characterize different fire stages including the smoldering phase (Kaiser et al., 2012). Recent studies have shown that fire emissions factors may vary widely depending on fire-behavior (van Leeuwen and van der Werf, 2011; Parker et al., 2016; Reisen et al., 2018), while improved knowledge of fire-climate interactions
are crucial for emissions forecasting (Di Giuseppe et al., 2018).

The Global Fire Atlas methodology builds on a range of previous studies that have used daily moderate resolution satellite imagery to estimate individual fire sizes (Archibald and Roy, 2009; Hantson et al., 2015; Frantz et al., 2016; Andela et al., 2017), shape (Nogueira et al., 2017; Laurent et al., 2018), duration (Frantz
et al., 2016) and spread dynamics (Loboda and Csiszar, 2007; Coen and Schroeder, 2013; Sá et al., 2017). We provide the first fire progression-based algorithm to map individual fires across all biomes, including the first global estimates of ignition locations and timing, duration, daily expansion, fire line, speed and direction of spread. Several previous studies have estimated fire size distributions based on a flood-fill algorithm, where all neighboring pixels within a certain time threshold are classified as the same fire
(Archibald and Roy, 2009; Hantson et al., 2015). Interestingly, we found similar spatial patterns of fire size (cf. Fig. 8 and Archibald et al., 2013; Hantson et al., 2015), although absolute estimates may show large differences based on the "cut off" value used within the flood-fill approach (Oom et al., 2016), and to a lesser extent by the fire persistence threshold used here. Spatial patterns of fire size and duration also compared favorably with estimates of Frantz et al. (2016) for southern Africa (Fig. 9a) and estimates of fire
speed by Loboda et al. (2007) for Central Asia (Fig. 10a). Here we compared our results to fire perimeter estimates from the MTBS (Eidenshink et al., 2007; Sparks et al., 2015). Moderate agreement was found for forested ecosystems and shrublands, but results differed more in grassland biomes (Fig. 5). Interestingly, we found that the poor agreement in grasslands stemmed from differences in the spatial and temporal resolution of the burned area estimates (Fig. B2). In line with previous studies, we found that the coarser
resolution (500 m) of the MODIS burned area data used to develop the Global Fire Atlas sometimes underestimated overall burned area (e.g. Randerson et al., 2012; Rodrigues et al., 2019; Roteta et al., 2019), fragmenting individual large fires. However, the Landsat-based MTBS data at 30 m resolution were unable to distinguish individual fires within large burn patches of fast-moving grassland fires based on infrequent Landsat satellite overpasses (Fig. B2).

An initial accuracy assessment of Global Fire Atlas fire perimeter estimates for the continental US revealed several important limitations and opportunities for further development of individual fire characterization using satellite burned area data. In addition to the accuracy assessment of fire perimeters, we also investigated the temporal accuracy of the Global Fire Atlas (Fig. 4) as well as the fire duration estimates
(Fig. 6) based on active fire detections. Low to moderate correlations ($r^2$ ranging from 0.3 to 0.5) were found between Global Fire Atlas and fire duration estimates based on a combination of MTBS fire perimeters and VIIRS active fire detections. Disagreement partly originated from differences in fire perimeter estimates as well as differences between the day-of-burn estimates derived from the MCD64A1 burned area data and VIIRS active fire detections. Moreover, the uncertainty in the burn date of the
underlying burned area product is typically at least one day, resulting in a large uncertainty in the fire duration estimates of shorter fires (Fig. 6). The temporal accuracy of the Global Fire Atlas adjusted burned area compared to VIIRS active fire detections ranged from 41% on the same day and 80% within ± 1 day

in shrublands to and 24% (same day) and 54% (± 1 day) in forests. However, in forested ecosystems the use of active fire detections for validation purposes is not ideal, as fires may smolder for days, triggering active fire detections after the fire front has passed. Understanding the temporal accuracy of the Global Fire Atlas products is important for linking individual fire dynamics to fire weather, and we found good agreement between Global Fire Atlas and US Forest Service fire expansion using a 3-day running average, but less good agreement for individual days based on burn date uncertainty (Fig. B1). Other parameters, including fire speed and direction of spread, were not validated during this stage. However, our comparison to daily fire perimeter estimates from the US Forest Service showed good agreement in terms of average expansion rates, suggesting reasonable overall estimates of speed (Fig. 7). Overall, there is a need to develop additional validation methodologies and data products to advance our understanding of satellite-derived estimates of individual fire behavior, building on the long-standing efforts for burned area (Boschetti et al., 2009) and active fires (Schroeder et al., 2008).

In addition to the Global Fire Atlas algorithm, the data quality also depends on the underlying global burned area product (MCD64A1 c6). In particular, several recent studies have shown that moderate resolution burned area products are unable to adequately map the occurrence of small fires (~ ≤100 ha) in the United States (Fusco et al., 2019) and savanna regions of Brazil (Rodrigues et al., 2019) and Africa (Roteta et al., 2019), resulting in a considerable underestimate of global burned area (Randerson et al., 2012; Giglio et al., 2018). Therefore, care should be taken when using the Global Fire Atlas for cropland regions or other regions dominated by small fires (see Fig. 8c). The quality of derived parameters in the Global Fire Atlas for these same regions also depends on the fire persistence threshold we used to identify when fires spread from one grid cell into the next. The thresholds we used may be more appropriate for analysis of fires in natural landscapes than in croplands with synchronized small fire activity across multiple adjacent fields. Finally, daily burned area products do not resolve the diurnal cycle of fire activity; fire lifetime and fire behavior may vary widely across fire regimes (Freeborn et al., 2011; Andela et al., 2015), and sub-daily fire dynamics cannot be resolved in the Global Fire Atlas. In line with these limitations, we found that Global Fire Atlas data performed best for large fires (Figs. 6 and 7). Further development of the Fire Atlas product suite is possible based on improvements in the underlying burned area data from multiple satellite sensors as well as new active fire products at higher spatial resolution (e.g., VIIRS). The Global Fire Atlas algorithm provides a flexible framework that can be easily adjusted to work at different spatial or temporal resolutions.

## 5 Data availability

The data are freely available at http://www.globalfiredata.org in standard data product formats and updates for subsequent years will be distributed pending availability of MCD64A1 burned area data and associated research funding. Global per-fire-year shapefiles of the ignition locations (point) and individual fire perimeters (polygon) contain attribute tables with a unique fire ID, ignition location, start and end dates, size, duration, and average values of the daily expansion, daily fire line, speed, and direction of spread (Fig. 1, Table A1). In addition, gridded 500 m global maps of the Global Fire Atlas adjusted burn dates, daily fire line, speed and direction of spread are available in GeoTIFF format. A monthly gridded GeoTIFF product is also available at 0.25° resolution. Global Fire Atlas data products can also be visualized and evaluated using an online tool at http://www.globalfiredata.org to explore individual fire characteristics for a selected region of interest.

## 6 Conclusions

The Global Fire Atlas is a new publicly available global dataset on seven key fire characteristics: ignition location and timing, fire size, duration, daily expansion, daily fire line, speed, and direction of spread. Over

the 2003 – 2016 study period, we identified 13,250,145 individual fires ($\geq$ 21 ha) based on the moderate resolution MCD64A1 collection 6 burned area data. Striking differences were observed among global fire regimes along gradients of ecosystem productivity and human land use. In general, in ecosystems of abundant fuel and low human influence, large fires of long duration dominated total burned area, with small fires contributing most to overall burned area in human-dominated regions or areas too wet for frequent fires. Fires moved quickly through arid ecosystems with low fuel densities but fire sizes were eventually limited by fuels from natural or human landscape fragmentation. The dataset enables new lines of investigation for understanding vegetation-fire feedbacks, climatic and human controls on global burned area, fire forecasting, emissions modeling, and benchmarking of global fire models.

**Appendix A: Supporting material for the methods**

**Table A1: Overview of the Global Fire Atlas data-layers.** The shapefiles of ignition locations (point) and fire perimeters (polygon) contain attribute tables with summary information for each individual fire, while the underlying 500 m gridded layers reflect the day-to-day behavior of the individual fires. In addition, we provide aggregated monthly layers at 0.25° resolution for regional and global analyses.

| | Shapefile attributes* | 500 m daily gridded | 0.25° monthly gridded |
|---|---|---|---|
| **Ignitions** | location and timing | - | sum |
| **Perimeter (km)** | per fire | - | - |
| **Size (km$^2$)** | per fire | - | average |
| **Duration (days)** | per fire | - | average |
| **Daily fire line (km)** | average per fire | yes | average |
| **Daily fire expansion (km$^2$ day$^{-1}$)** | average per fire | - | average |
| **Speed (km day$^{-1}$)** | average per fire | yes | average |
| **Direction of spread (-)** | dominant per fire | yes | dominant |
| **Day of burn** | - | yes | - |

* vector data are derived from the underlying 500 m MODIS data.

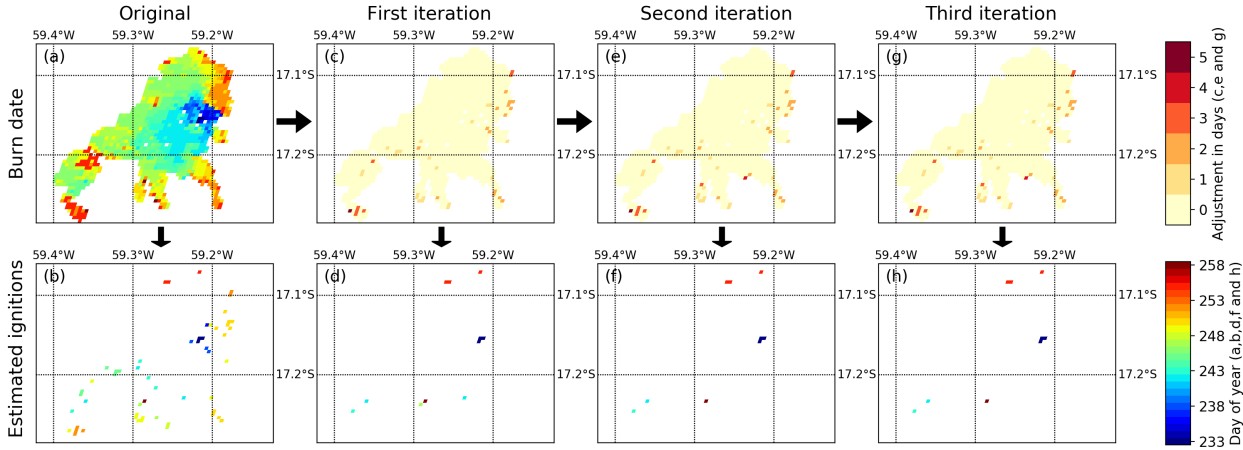

**Figure A1: Burn date adjustment to remove local minima that are not associated with ignition points.** (a) MCD64A1 burn date estimate for the 2015 example fires in the Brazilian Cerrado ecosystem, (b) local minima within (a). (c) Burn date adjustment after the first iteration, and (d) resulting local minima. (e) Burn date adjustment after the second iteration, and (f) resulting local minima. (g) Burn date adjustment after the third iteration, and (h) resulting local minima. Note that for these particular fires there was no difference between (e and f) and (g and h), and the final iteration has no added value here. We found that multiple iterations were particularly beneficial for slow moving fires in forested ecosystems.

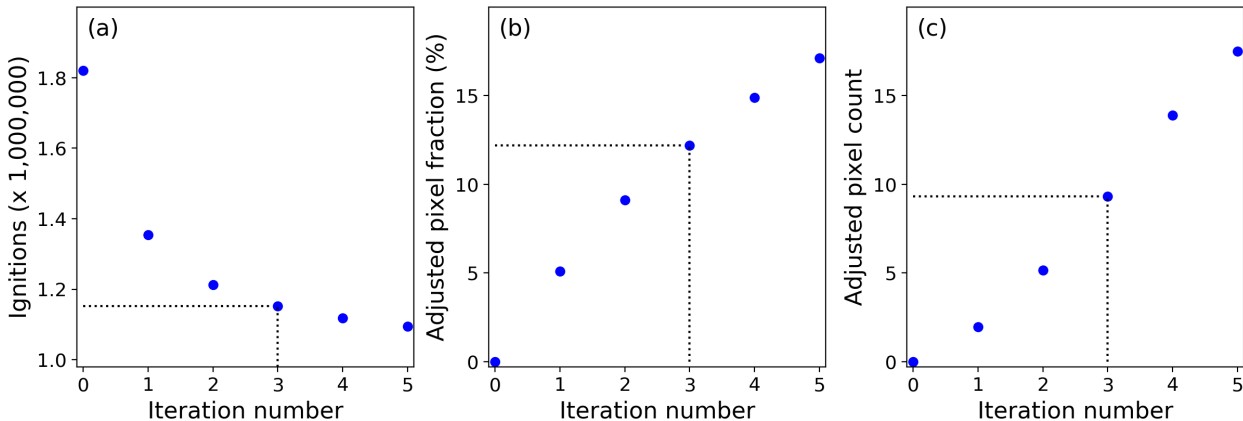

**Figure A2: Tradeoffs between reducing local minima not associated with ignition locations and adjustments made to the global burned area product.** (a) Local minima (ignitions) detected within the daily 500 m global burned area data for 2015 after different number of iterations of the ignition point filter, (b) corresponding fraction of burned area pixels with adjusted burn date, and (c) corresponding number of burned area pixels adjusted divided by the reduction in ignition count. In this study, we used three iterations of the ignition point filter (indicated with the intermittent lines in figures a, b and c), and "0 iterations" refers to the original MCD64A1 col. 6 burned area data.

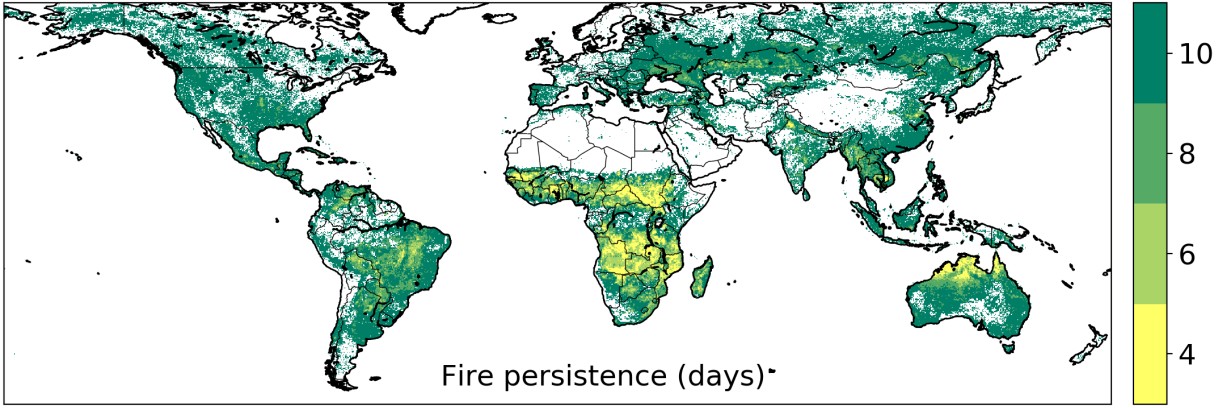

**Figure A3: Average fire persistence threshold at 0.25° resolution.** The fire persistence threshold determines how long a fire may take to spread from one 500 m grid cell into the next. We used a 4-day fire persistence threshold for 500 m grid cells that burned more than 3 times during the study period (2003 - 2016), and a 6, 8 and 10-day fire persistence period for grid cells that burned 3 times, 2 times, or 1 time, respectively.

## Appendix B: Supporting material for the results and discussion

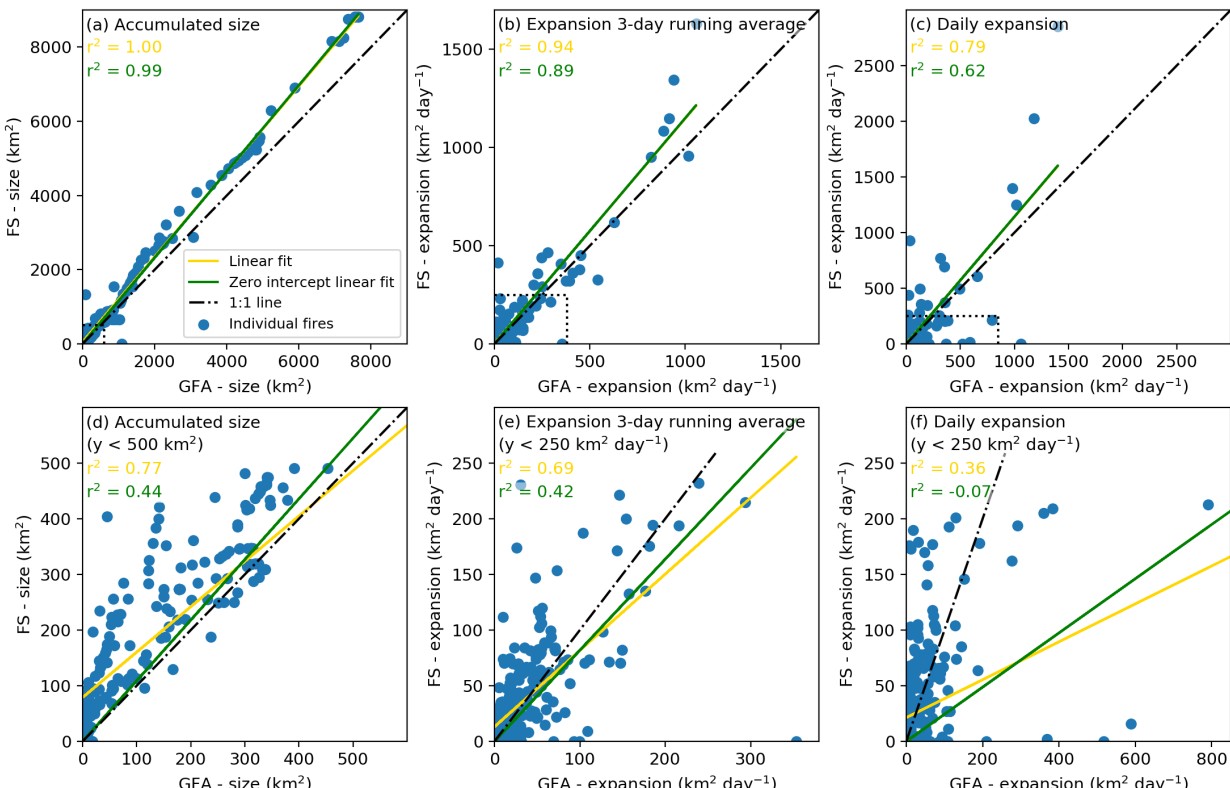

**Figure B1: Comparison of daily Global Fire Atlas and US Forest Service data for a selected number of well characterized wildfires in the US.** (a) The accumulated daily fire size (for all fires, N=15) illustrates the ability of the Global Fire Atlas to reproduce individual large fire sizes at any specific day over the fire lifetime (each blue dot indicates the size of a specific fire on a specific day). (b) A 3-day running average of the daily growth or "expansion" of each fire (km$^2$ day$^{-1}$) and (c) the daily expansion on each day of each fire. Figures (d), (e), and (f) are like (a), (b), and (c), but for US Forest Service fire sizes smaller than 500 km$^2$ or expansion rates lower than 250 km$^2$ day$^{-1}$ and corresponding Global Fire Atlas estimates (see intermittent boxes on top-figures).

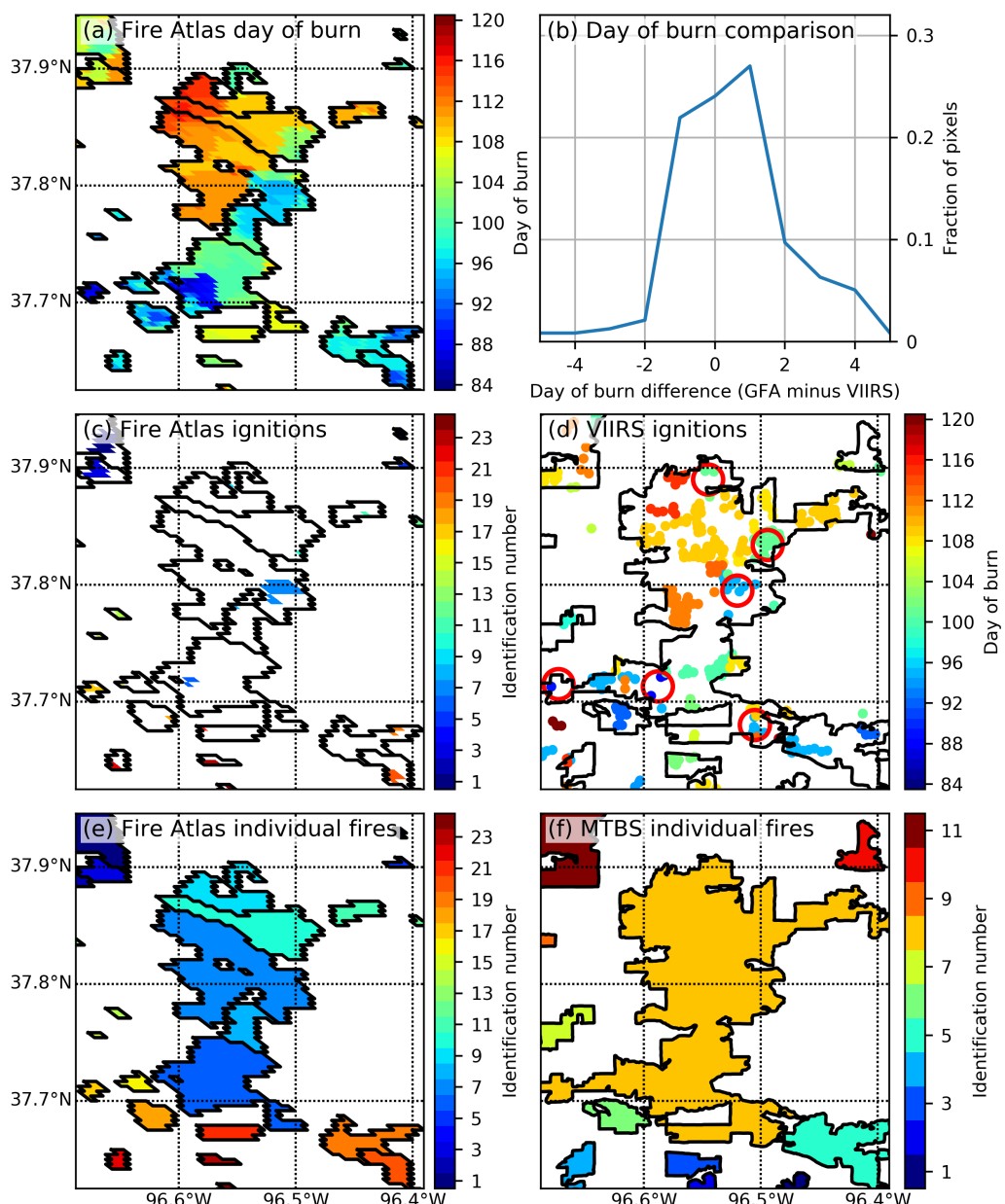

**Figure B2: Comparison of Global Fire Atlas perimeters and ignition locations to estimates based on MTBS and VIIRS for frequently-burning grasslands in Kansas, USA.** (a) Global Fire Atlas adjusted burn dates from MCD64A1, (b) per-pixel comparison of adjusted burn dates used within the Global Fire Atlas (GFA) to the day of the active fire detection from VIIRS, (c) ignition points as estimated by the Global Fire Atlas, (d) manually interpreted ignition locations (red circles) based on VIIRS active fire detections on top of MTBS fire perimeters, (e) individual fires as estimated by the Global Fire Atlas, and (f) the MTBS burned area and individual fires. Here, MCD64A1 data underestimated the total burned area compared to the visual interpretation of Landsat data within the MTBS project, resulting in fragmentation of individual large fires. However, the daily temporal resolution of MODIS imagery allowed the Global Fire Atlas to distinguish individual fires and ignition points within larger burn scars that cannot be resolved from infrequent Landsat observations used to delineate fire perimeters within the MTBS project. Broad patterns of ignition locations identified by the Global Fire Atlas were confirmed by manual interpretation of patterns inferred from VIIRS active fire detections (d).

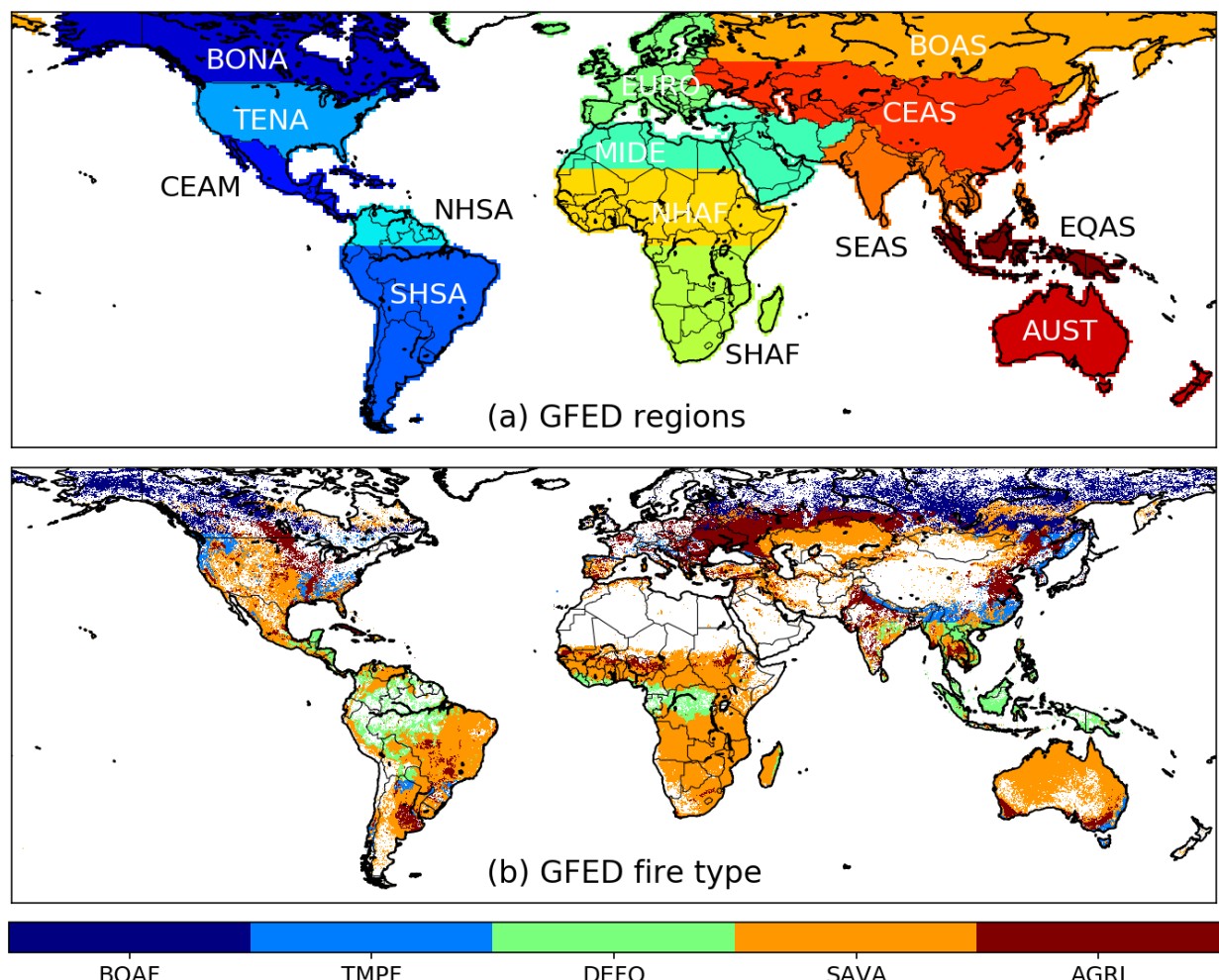

**Figure B3: Global Fire Emissions Database (GFED) regions and dominant GFED fire types used for Tables 1 and 2.** (a) GFED regions used in Table 1, and (b) GFED dominant fire type as used in Table 2. Abbreviations of the GFED regions shown in (a) are: boreal North America (BONA), temperate North America (TENA), Central America (CEAM), northern hemisphere South America (NHSA), southern hemisphere South America (SHSA), Europe (EURO), Middle East (MIDE), northern hemisphere Africa (NHAF), southern hemisphere Africa (SHAF), boreal Asia (BOAS), Central Asia (CEAS) southeast Asia (SEAS), equatorial Asia (EQAS), and Australia and New Zealand (AUST). Abbreviations of the GFED fire types shown in (b) are: boreal forest (BOAF), Temperate forest (TMPF), Tropical forest deforestation (DEFO), savanna (SAVA) and agriculture (AGRI).

**Author contributions.** NA, DCM, and JTR designed the study. NA carried out the data processing and analysis. All authors contributed to the interpretation of the results and writing of the manuscript.

**Competing interests.** The authors declare that they have no conflict of interest.

**Acknowledgements.** This work was supported by NASA's Carbon Monitoring System program (grant 80NSSC18K0179) and the Gordon and Betty Moore Foundation (grant GBMF3269). We thank Thomas Mellin of the US Forest Service for granting access to the daily fire perimeter collected over the western USA.

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
