# Peer review of "The Global Fire Atlas of individual fire size, duration, speed, and direction"

_Earth System Science Data, 2018_

## Referee Comment (RC1) · Anonymous Referee #1 · 10 Oct 2018

**General comment**

Andela et al., present a novel and very important dataset of several important fire characteristics globally on a daily basis. This dataset will serve earth system and social scientists on topics ranging from (but not limited to) fire emission estimates in earth system modelling, feedback between fires and ecosystem, fire management and studies of socio-economic feedback of fires. The manuscript is well written and the underlying methodologies have been explained precisely. Being the first dataset of such kind, a complete validation was challenging. However the authors have used the available resources, e.g. VIIRS (globally for four different ecosystems) for burn date, MTBS in the US for the fire perimeter and a combination of both for the fire duration.

The dataset, however, has a large uncertainty for short fires (persisting for less than a day, for example, crop residue fires), which is acknowledged in the discussion. I have only minor comments regarding this manuscript and recommend publication of this manuscript in ESSD after the authors have addressed them:

**Specific comments:**

The methodology considers clusters of fires in a given fire season (12 months) as a starting point. What if the fires season is less than 12 months? For example, the same area is burnt twice after a gap of six months? As per my understanding, the local minima filter will only assign it to the later burnt date of the fire season. This will also have consequences on the estimation of fire duration and perimeter.

The authors conclude that this dataset is useful for emission modelling. In my opinion, the authors should also acknowledge the limitation of this dataset for use in atmospheric models for emission estimates from fires. The Global Fire Atlas does not take into account the smoldering stage of fires, which significantly contribute to gas and particle emissions. In this context, the work of Kaiser et al., 2011 should be mentioned, which uses the fire radiative power for emission estimates.

Kaiser, J. W., et al. (2012), Biomass burning emissions estimated with a global fire assimilation system based on observed fire radiative power, Biogeosciences, 9(1), 527-554, doi:10.5194/bg-9-527-2012.

**Page 4, line 155:** What fraction of local minima is discarded after each iteration step? This information is important for optimization of the number of iteration (which was taken to be 3 in the present work).

**Figure 4:** The horizontal axis legend ($\Delta$ burn date (burned area minus active fires)) is not clear to me.

**Figure 7:** Please check the units in the middle panel (for ignitions).

The discussion regarding fire direction on page 14 is relatively weak. The fire directions are highly variable depending on topographical features, prevalent wind field and fuel availability. What can one conclude from such variable fire direction and how this information is useful?

The Global Fire Atlas dataset is available for the year 2003-2016. Will this dataset be continuously updated? Given that the dataset is so important, the authors should provide information of update frequency and policy.

---

## Referee Comment (RC2) · Anonymous Referee #2 · 23 Nov 2018

The paper is relevant as it tries to provide a new approach to the analysis of fire regimes, by analyzing different parameters of individual fires extracted from global burned area products. This effort is relevant to better parameterize fire models, as well as to understand fire trends affected by changing climate and socio-economic conditions. The main problem I found in this paper is their ambition to qualify single fire activity from a product that was not derived from this purpose. Recent papers (Padilla et al., 2015; Padilla et al., 2014) have found that global burned area products have important omission and commission errors, particularly for small fires Chuvieco et al., 2018; Roteta et al., 2018. They provide a good image of fire activity at global scale, meanwhile the analysis is done at global or at much continental scale. However, establishing characteristics of single fires from these products may be quite misleading. If the

authors do not provide better validation datasets, the parameters they analyze at global scale may be in fact confusing. In my view, this is the main weakness of the paper. The authors are assuming estimations from a dataset that is not really validated. Until the MCD64A1 is fully validated, and we better understand their strengths and weaknesses, deriving such detailed analysis as presented in this paper may create more confusion than knowledge. In fact the comparison (validation is not an adequate term for what the authors include in the manuscript) analysis show a high degree of uncertainty even for the simplest variable (fire perimeter). When perimeters are compared with those derived from higher resolution data (MTBS), the correlations are low (for the authors, line 578: they are "reasonable correlations (r2 ranging from 0.3 to 0.5)", but we should remember that they imply than 70-50% of the variance is unexplained). Therefore, in my opinion the subsequent analyses derived from this dataset are quite likely to be erroneous. The comparison they made with active fires and MTBS shows also poor agreements in all biomes. What about fire speed or direction? I suggest that they at least compare their results with specific very large fires where fire growth is available for different forest services, to check if at least for those large fires their estimations are correct. Very large fires could also be assessed using Landsat data, at least for fire perimeter-size and shape. Are you sure that Australia had a single fire of 42.000 km2? They could also compare their outputs with models of global fire weather conditions (Jolly et al., 2015; Pettinari and Chuvieco, 2017), as well as include some comparisons with fire spread and duration published by fire behavior experts. On the other hand, I doubt about the utility of providing global averages of different fire parameters, such as fire duration or progression by continent. In this regard, some of the comments included in the results section may seem quite trivial or difficult to justify empirically. What is the point of concluding that "fire duration exerted a strong control on fire size and total burned area"? Is this not the case in the vast majority of fires? In summary, the authors should make an additional effort to really validate their product and better identify the weaknesses of current analysis.

Specific comments Line 45: Worldwide, fires burn an area larger than the size of the

[Figure]

European Union every year (Randerson et al.,2012; Giglio et al., 2013). Please include total area in km2, the reader does not need to know the size of the European union to understand your sentence. Line 55: you claim that burned area reduction is occurring in the last two decades, but Andela et al., 2017 paper refers only to the 2001-2017 period (1995-2001 with more uncertainty), so you could only claim that the reduction is observed in the last few years, as you do not have date from several decades ago. Line 65: Our understanding of global fire activity is also severely constrained by the coarse resolution data we are based on our analysis. Recent analysis of burned area estimation comparing coarse and medium resolution data shows that in fact we may be losing a significant part of fire activity (Roteta et al., 2018, https://geogra.uah.es/fire_cci/sfd.php), particularly in tropical regions. Line 88: update (Giglio et al., submitted) Lines 155-164: How did you proceed in the case of small fires (a few pixels)? You claim that local minima are deleted when they do not spread forward in time. Lines 180-187: Fire spread is obviously associated to wind speed and slope, not just to fuel availability. Therefore the assumptions made by the authors seem quite arbitrary for a global product. Have they made any validation of their persistence algorithm? It is not clear what happened with areas that burned 2 times, were they assigned 6 or 8 day persistency? The thresholds are in fact overlapped. Line 195. It is not clear if two active fires that merged were assigned a single perimeter or two. It seems they were divided, but most forest services would probably consider them as single one. Lines 240-... It is not clear what the authors did when areas were not observed by clouds or cloud shadows. What is the impact of unobserved periods in fire progression? Were the geometrical deformation effects caused by off-nadir observations taken into account? Figure 3 shows direction of spread that are not very realistic, as all sort of directions are included, even for neighbor pixels (North and South directions in contiguous areas??) It is not clear why did you include MCD64 in Figure 4, as the date information should be the almost the same as the Global Fire Atlas. I would recommend changing it to a single graph showing dating accuracy for the four major biomes The fire dominant direction will probably be more useful for fire modelers ex-

pressed in degrees. Other authors have done similar analysis, a recent one by Laurent et al., 2018 Line 440. I doubt that any fire behavior modeler would agree with: "... the dominant direction typically represented less than half of the pixels". I think the approach by Laurent et al (2018) using the dominant direction of the evolving ellipsis is more adequate in this regard, as most fires have a dominant wind direction. I do not understand the meaning of using average NDVI values to show extreme fires. I do not see the relation.

References Chuvieco, E., Lizundia-Loiola, J., Pettinari, M. L., Ramo, R., Padilla, M., Tansey, K., Mouillot, F., Laurent, P., Storm, T., Heil, A., and Plummer, S.: Generation and analysis of a new global burned area product based on MODIS 250 m reflectance bands and thermal anomalies, Earth Systems Science Data, 2018, 2015-2031, Doi: https://doi.org/10.5194/essd-10-2015-2018, 2018. Jolly, W. M., Cochrane, M. A., Free-born, P. H., Holden, Z. A., Brown, T. J., Williamson, G. J., and Bowman, D. M.: Climate-induced variations in global wildfire danger from 1979 to 2013, Nature Communications, 6, Doi: 10.1038/ncomms8537, 2015. Laurent, P., Mouillot, F., Yue, C., Ciais, P., Moreno, M. V., and Nogueira, J. M. P.: FRY, a global database of fire patch functional traits derived from space-borne burned area products, Scientific Data, 5, 180132, Doi: 10.1038/sdata.2018.132, 2018. Padilla, M., Stehman, S. V., and Chuvieco, E.: Validation of the 2008 MODIS-MCD45 global burned area product using stratified random sampling, RSE, 144, 187-196, Doi: http://dx.doi.org/10.1016/j.rse.2014.01.008, 2014. Padilla, M., Stehman, S. V., Hantson, S., Oliva, P., Alonso-Canas, I., Bradley, A., Tansey, K., Mota, B., Pereira, J. M., and Chuvieco, E.: Comparing the Accuracies of Remote Sensing Global Burned Area Products using Stratified Random Sampling and Estimation, RSE, 160, 114-121, Doi: http://dx.doi.org/10.1016/j.rse.2014.01.008, 2015. Pettinari, M., and Chuvieco, E.: Fire Behavior Simulation from Global Fuel and Climatic Information, Forests, 8, 179, 2017. Roteta, E., Bastarrika, A., Storm, T., and Chuvieco, E.: Development of a Sentinel-2 burned area algorithm: generation of a small fire database for northern hemisphere tropical Africa RSE, (in review), 2018.

---

## Author Comment (AC1) · 23 Jan 2019

General comment

Andela et al., present a novel and very important dataset of several important fire characteristics globally on a daily basis. This dataset will serve earth system and social scientists on topics ranging from (but not limited to) fire emission estimates in earth system modelling, feedback between fires and ecosystem, fire management and studies of socio-economic feedback of fires. The manuscript is well written and the underlying methodologies have been explained precisely. Being the first dataset of such kind, a complete validation was challenging. However the authors have used the available resources, e.g. VIIRS (globally for four different ecosystems) for burn date, MTBS in the US for the fire perimeter and a combination of both for the fire duration.

The dataset, however, has a large uncertainty for short fires (persisting for less than a day, for example, crop residue fires), which is acknowledged in the discussion. I have only minor comments regarding this manuscript and recommend publication of this manuscript in ESSD after the authors have addressed them:

We thank the reviewer for his/her constructive comments and thoughtful review. Please find our detailed response along with the suggested changes to our manuscript below. Note that we will upload the updated manuscript using track change (in response to both reviews) in a separate post.

Specific comments:

The methodology considers clusters of fires in a given fire season (12 months) as a starting point. What if the fires season is less than 12 months? For example, the same area is burnt twice after a gap of six months? As per my understanding, the local minima filter will only assign it to the later burnt date of the fire season. This will also have consequences on the estimation of fire duration and perimeter.

This is correct, we try to minimize the amount of pixels that burned twice during a single burning season by defining the burning season as "5 months before until 6 months after the month of maximum mean burned area" for each individual 10° x 10° MODIS tile. In most of the world (particularly areas that burn frequently) the fire season is quite clearly defined, e.g. wet and dry seasons in the tropics or cold winters and warm summers at higher latitudes; however, in regions without clear seasonality (e.g. always dry or wet), or some areas with both natural and cropland fires, our methodology is not ideal. In case there was overlap between two burning events we only retain the earliest burn dates. Therefore, a small fraction (<1%) of global burned area is effectively removed from our dataset, indeed affecting fire perimeters by reducing overall burned area. The advantage of our methodology is that we can produce user friendly global "annual" layers of fire behavior, both gridded at 500-m resolution, as well as in the form of shapefiles.

In response to this suggestion will more clearly explain these tradeoffs. In particular, we will rephrase lines 134-135 to: "This approach results in a small reduction of total burned area, but allows us to produce user friendly global annual layers in both gridded and shapefile format."

The authors conclude that this dataset is useful for emission modelling. In my opinion, the authors should also acknowledge the limitation of this dataset for use in atmospheric models for emission estimates from fires. The Global Fire Atlas does not take into account the smoldering stage of fires, which significantly contribute to gas and particle emissions. In this context, the work of Kaiser et al., 2011 should be mentioned, which uses the fire radiative power for emission estimates. Kaiser, J. W., et al. (2012), Biomass burning emissions estimated with a global fire assimilation system based on observed fire radiative power, Biogeosciences, 9(1), 527-554, doi:10.5194/bg-9-527-2012.

We fully agree with this suggestion, although our estimates of fire behavior may provide some first guidance on where smoldering may occur (e.g. slow multi-day fires), or where fires may burn more intensely (e.g. high speed), this is further modified by e.g. fuel loads and conditions. Moreover, it often remains unclear how the combination of fire behavior and fuels modify emissions factors (i.e. composition of emissions), and thus eventual emissions of different trace gasses and aerosols.

In the updated manuscript, we will discuss this in more detail. In particular, we will change lines 546-547 to "Large differences in fire behavior across ecosystems and management strategies may improve fire emissions estimates and emissions forecasting, particularly when combined with active fire detections to better characterize different fire stages including the smoldering phase (Kaiser et al., 2012)."

Page 4, line 155: What fraction of local minima is discarded after each iteration step? This information is important for optimization of the number of iteration (which was taken to be 3 in the present work).

During our development phase we had looked into this for a number of individual MODIS tiles, and found that 3 iterations may provide an optimal threshold across different ecosystems. We also found that forest fires may generally require more iterations than fast-moving grassland fires. In the updated manuscript we will include a new supplementary figure visualizing these tradeoffs, to support our decision of 3 iterations (Fig. 1 here).

[Figure]

Figure 1 (new Fig. A2 in manuscript): **Tradeoffs between reducing local minima not associated with ignition locations and adjustments made to the global burned area product.** (a) Local minima (ignitions) detected within the daily 500 m global burned area data for 2015 after different number of iterations of the ignition point filter, (b) corresponding fraction of burned area pixels with adjusted burn date, and (c) corresponding number of burned area pixels adjusted divided by the reduction in ignition count. In this study, we used three iterations of the ignition point filter (indicated with the intermittent lines in figures a, b and c), and "0 iterations" refers to the original MCD64A1 col. 6 burned area data.

Figure 4: The horizontal axis legend ( burn date (burned area minus active fires)) is not clear to me.

The horizontal axis indicates the difference in burn date between VIIRS active fire detections and the burned area datasets (MCD64A1 c6 and the adjusted burned area data by the Global Fire Atlas). This is calculated as the burn date of the burned area data minus the associated burn date of the (first) corresponding active fire detection. Thus, a negative number indicates that the burned area was detected before the active fire detection, zero indicates a perfect match, and a positive number indicates that the burned area was detected later than the first active fire detection.

We will change the x-axis label to "Difference in day of burn compared to VIIRS (days)" and change the y-axis label to "Pixel fraction".

Then, we will change the figure caption to: "**Per pixel global comparison of burn dates derived from the MCD64A1 burned area product, adjusted burn dates of the Global Fire Atlas, and VIIRS active fire detections (2012 – 2016).** (a) Forests, (b) shrublands, (c) woody savannas, and (d) savannas and grasslands. Negative values indicate pixels with a burned area day of burn earlier than the first corresponding VIIRS active fire detection, zero indicates no difference in day of burn between both datasets, and positive numbers indicate a delayed detection of burned area compared to active fire detections."

Figure 7: Please check the units in the middle panel (for ignitions).

Although we believe the units are correct, we appreciate that the units on this figure may be somewhat confusing. In particular because we state that burned area is the product of ignitions and size. We think the confusion arises because the exact surface area of a 0.25° grid cell varies with latitude, therefore we feel that for ignitions it makes most sense to report the "ignition density" per unit of area per year. In a similar fashion we report burned area as a fraction per year rather than in square kilometers per year.

For clarity, we will change the figure label "(b) Ignitions ($km^{-2}$ $yr^{-1}$)" to "(b) Ignition density ($km^{-2}$ $yr^{-1}$)". Also, we will further clarify this in the figure caption: "**Figure 8: Average global burned area (MCD64A1), ignition density, and fire size over the study period 2003 – 2016.** For any given area (a) burned area in $km^2$ per year would be the product of (b) ignitions per year and (c) fire size in $km^2$. However, because the size of a 0.25° grid cell varies with latitude we have converted the units of burned area to fraction (%) per year and of ignitions to number per $km^2$ per year for spatial consistency."

The discussion regarding fire direction on page 14 is relatively weak. The fire directions are highly variable depending on topographical features, prevalent wind field and fuel availability. What can one conclude from such variable fire direction and how this information is useful?

We had also anticipated a stronger effect of the dominant wind direction. Therefore, we think that variability in fire direction is an interesting finding on its own. As we show, landscape features and other factors play an important role in fire spread direction, leading to heterogeneous patterns of fire spread in all biomes. This finding may help improve global fire models, for example, since models often assume that fire growth can be described by relatively simple growth equations with homogenous fuel beds. Our work adds to an increasing body of evidence that landscape heterogeneity and associated variability in fuel conditions have a strong influence on global fire behavior across scales.

We will add an additional sentence to the discussion section to highlight to potential new insights for fire modeling (line 546):
"In a similar fashion, many models assume relatively homogeneous fuel beds, while our results suggest that landscape features and vegetation patterns result in highly heterogeneous fuel beds that form a strong control on fire spread (speed and direction)."

The Global Fire Atlas dataset is available for the year 2003-2016. Will this dataset be continuously updated? Given that the dataset is so important, the authors should provide information of update frequency and policy.

We aim to update the dataset annually, with a delay of about 1 year. Because of the "per fire year" processing the algorithm requires burned area data up to 6 months after the calendar year ends to process a given year while the burned area product (MCD64A1 col. 6) is also released with a few months delay.

We will also mention this in the "Data availability" section: "The data are freely available at http://www.globalfiredata.org in standard data product formats and updates for subsequent years will be distributed pending availability of MCD64A1 burned area data and associated research funding."

Reference:

Kaiser, J. W., Heil, A., Andreae, M. O., Benedetti, A., Chubarova, N., Jones, L., Morcrette, J. J., Razinger, M., Schultz, M. G., Suttie, M. and van der Werf, G. R.: Biomass burning emissions estimated with a global fire assimilation system based on observed fire radiative power, Biogeosciences, 9(1), 527–554, doi:10.5194/bg-9-527-2012, 2012.

---

## Author Comment (AC2) · 23 Jan 2019

The paper is relevant as it tries to provide a new approach to the analysis of fire regimes, by analyzing different parameters of individual fires extracted from global burned area products. This effort is relevant to better parameterize fire models, as well as to understand fire trends affected by changing climate and socio-economic conditions.

We thank the reviewer for his/her review and thoughtful suggestions. Please find a detailed response to the individual suggestion along with proposed changes below. Note that we will upload the updated manuscript using track change (in response to both reviews) in a separate post.

The main problem I found in this paper is their ambition to qualify single fire activity from a product that was not derived from this purpose. Recent papers (Padilla et al., 2015; Padilla et al., 2014) have found that global burned area products have important omission and commission errors, particularly for small fires Chuvieco et al., 2018; Roteta et al., 2018. They provide a good image of fire activity at global scale, meanwhile the analysis is done at global or at much continental scale. However, establishing characteristics of single fires from these products may be quite misleading. If the authors do not provide better validation datasets, the parameters they analyze at global scale may be in fact confusing. In my view, this is the main weakness of the paper. The authors are assuming estimations from a dataset that is not really validated. Until the MCD64A1 is fully validated, and we better understand their strengths and weaknesses, deriving such detailed analysis as presented in this paper may create more confusion than knowledge.

We appreciate this suggestion, and are aware of the shortcomings of moderate resolution (500-m) satellite imagery (e.g. omission of small fires). Unfortunately, the high resolution satellite data (e.g. Landsat or Sentinel-2) and derived products do not provide the temporal accuracy required to track individual fires and their behavior. In response to this comment, we would like to make the following clarifications. First, the use of moderate resolution satellite imagery to track individual wildfire behavior is an already widely used concept (e.g. Loboda and Csiszar, 2007; Archibald and Roy, 2009; Veraverbeke et al., 2014; Hantson et al., 2015; Benali et al., 2016; Frantz et al., 2016; Fusco et al., 2016; Nogueira et al., 2016; Oom et al., 2016; Laurent et al., 2018). Building on these previous studies, our manuscript provides an improved global approach to identify individual fires and characterize their behavior based on an algorithm that identifies ignition locations and then tracks how the fire expands through time. Second, our aim was to develop a flexible algorithm that leverages availability of daily satellite observations at moderate resolution but can be applied easily to other (global) daily burned area data sets. The MCD64A1 col. 6 burned area product (succeeding MCD45 and MCD64A1 col. 5) is currently among the most widely used and best performing global burned area products (e.g. Padilla et al., 2015; Giglio et al., 2018; Humber et al., 2018), hence our choice for this data set. The MCD64A1 col. 6 data has now been officially released by NASA and was Stage-2 validated against 108 Landsat scenes (Giglio et al., 2018), and although we are looking forward to see additional validation, we see no reason why the data should not be used in the interim. Given the aim of our work (i.e. to develop a flexible algorithm to track individual fire behavior in daily global burned area products), we focus on the quality of the derived products (e.g. burn date accuracy), rather than on absolute burned area (e.g. omission of small fires), although clearly many of these aspects are not entirely independent. During the coming years, we are looking forward continue to develop our algorithm and apply it to the latest generation of improved daily burned area products, e.g. from VIIRS. Moreover, there would be no reason why our algorithm could not be applied to high resolution (20/30m) satellite data, if (close to) a near daily revisiting time would be achieved within the next decade or so.

In response to this suggestion we will include additional validation data (see detailed response below) and we will make several textual clarifications to more extensively discuss previous work, highlight the objectives of our paper, and the dependency of our "derived" product on the underlying burned area data. Specifically, we will make the following textual changes:

Line 107 "The Global Fire Atlas algorithm can be applied to any moderate resolution daily global burned area product, and the quality of the resulting dataset depends both on the Fire Atlas algorithm as well as the underlying burned area estimates. Here we applied the algorithm to the MCD64A1 collection 6 burned area dataset (Giglio et al., 2018) and the minimum detected fire size is therefore one MODIS pixel (21 ha). Several studies have shown that the MCD64A1 col. 6 burned area product provides a considerable improvement compared to previous generation of moderate resolution global burned area products (Padilla et al., 2015; Giglio et al., 2018; Humber et al., 2018)."

Line 552: "The Global Fire Atlas methodology builds on a range of previous studies that have used daily moderate resolution satellite imagery to estimate individual fire sizes (Archibald and Roy, 2009; Hantson et al., 2015; Frantz et al., 2016; Andela et al., 2017), shape (Nogueira et al., 2017; Laurent et al., 2018), duration (Frantz et al., 2016) and spread dynamics (Loboda and Csiszar, 2007; Coen and Schroeder, 2013; Sá et al., 2017)."

Line 568: "In line with previous studies, we found that the coarser resolution (500 m) of the MODIS burned area data used to develop the Global Fire Atlas sometimes underestimated overall burned area (e.g. Randerson et al., 2012; Roteta et al., 2019), fragmenting individual large fires. However, the Landsat-based MTBS data at 30 m resolution were unable to distinguish individual fires within large burn patches of fast-moving grassland fires based on infrequent Landsat satellite overpasses (Fig. B2)."

Line 604: "The Global Fire Atlas algorithm provides a flexible framework that can be easily adjusted to work at different spatial and/or temporal resolutions."

In fact the comparison (validation is not an adequate term for what the authors include in the manuscript) analysis show a high degree of uncertainty even for the simplest variable (fire perimeter). When perimeters are compared with those derived from higher resolution data (MTBS), the correlations are low (for the authors, line 578: they are "reasonable correlations (r2 ranging from 0.3 to 0.5)", but we should remember that they imply than 70-50% of the variance is unexplained). Therefore, in my opinion the subsequent analyses derived from this dataset are quite likely to be erroneous. The comparison they made with active fires and MTBS shows also poor agreements in all biomes. What about fire speed or direction?
I suggest that they at least compare their results with specific very large fires where fire growth is available for different forest services, to check if at least for those large fires their estimations are correct. Very large fires could also be assessed using Landsat data, at least for fire perimeter-size and shape. Are you sure that Australia had a single fire of 42.000 km2? They could also compare their outputs with models of global fire weather conditions (Jolly et al., 2015; Pettinari and Chuvieco, 2017), as well as include some comparisons with fire spread and duration published by fire behavior experts.

The numbers ("$r^2$ ranging from 0.3 to 0.5") refer to the fire duration estimates that have higher uncertainty than perimeters (read line 578: "Reasonable correlations ($r^2$ ranging from 0.3 to 0.5) were found between Global Fire Atlas and fire duration estimates .."), that show an average $r^2$ of 0.51 across land cover types. We appreciate that much of the variance remains unexplained, but we are encouraged by these results. For example, although Landsat-based MTBS provides better estimates of overall burned area, the underlying data lack the temporal revisit frequency to identify individual fires in low biomass ecosystems where fires are typically short and move fast. As we will show in our new supplementary figure (Fig. 1 here), the Global Fire Atlas clearly outperforms MTBS in terms of identifying individual ignition locations, which explains why the $r^2$ values of fire perimeters drop from 0.65 in forests to 0.38 for grasslands (a similar decline in agreement was found for fire duration, with $r^2$=0.51 for forest and $r^2$=0.33 for grasslands). This is an important finding on its own, since MTBS is a widely used dataset. Because the uncertainty arises both from the Global Fire Atlas and the (combined) MTBS and VIIRS datasets, the use of least square

regression is in fact not representative for estimating data quality, we therefore also use orthogonal distance regression that accommodates uncertainties in both datasets and shows better overall agreement.

Although we agree that an extensive comparison to daily fire perimeters would be a great form of validation for day-of-burn, fire duration, expansion rates and final perimeter, these data are unfortunately not available at the ease and scale that the reviewer suggests. In response to this suggestion we have requested available data from the US Forest Service and manually compiled a small dataset consisting of 15 fires that were reasonably well documented (this is not the case for the majority of fires). In line with good agreement between Global Fire Atlas and MTBS estimates of fire perimeters in forested ecosystems (Figs. 5 and 6 in manuscript), very good agreement was found between Global Fire Atlas estimates and US Forest Service estimates of fire size ($km^2$; $r^2=1.00$), duration (days; $r^2=0.87$), and daily expansion ($km2\ day^{-1}$; $r^2=0.97$; see Fig. 2 here). The comparison also highlights some of the shortcomings of the Global Fire Atlas data that we already discuss; for example, we observed that the Global Fire Atlas tended to somewhat underestimate fire size and overestimate (small) fire duration, resulting in conservative estimates of fire expansion.

These data also allowed us to explore how well the Global Fire Atlas characterizes fire growth dynamics (Fig. 3 here). We find very good agreement between Global Fire Atlas and US Forest Service estimates of fire size at any specific point in time ($r^2=1.00$), good agreement between a 3-day running average of fire expansion rates from both sources ($r^2=0.94$), and somewhat reduced agreement for daily expansion rates from both sources ($r^2=0.79$). This reduced performance for daily estimates originates from the considerable uncertainty in the exact burn date in the burned area product (see Fig. 4 in manuscript), and thus the attribution of fire expansion rates to a specific day. In addition, we find that the Global Fire Atlas data compares particularly well for large fires or expansion rates, with lower $r^2$ values for smaller fires and expansion rates (e.g. compare upper and lower panels of Figs. 2 and 3 here). The combination of very precise fire perimeter maps from the US Forest Service and the focus on large fires, likely explains why the Global Fire Atlas shows better agreement in Fig. 2 here compared to Fig. 5 in the manuscript. Therefore we expect that extremely large fires, like the fire in Australia the reviewer mentions, are among the fires that are best captured by the Global Fire Atlas data. Large fires are generally well mapped by moderate resolution burned area algorithms (e.g. Fusco et al., 2019) as well as easy to characterize from the Global Fire Atlas perspective.

To respond to the specific comment concerning the suggestion of comparing our estimated daily fire behavior to fire weather indices, we have great interest in this, and it is something we are currently working on in a separate manuscript.

In addition to the new figures (Fig. 1-3 here), we will make a number of textual additions/clarifications:

Line 276: "Finally, we compared Global Fire Atlas data to a small (manually compiled) dataset of daily fire perimeters from the US Forest Service."

Line 318: "For specific large wildfires across the western USA, the US Forest Service National Infrared Operations (NIROPS; https://fsapps.nwcg.gov/nirops/) derives estimates of daily fire perimeters for fire management purposes by collecting night-time high resolution infrared imagery. This imagery is manually analyzed by trained specialists to extract the active fire front. Although these data provide a wealth of information, only few fires were completely and precisely documented. From their database we were able to extract 15 large fires for which daily perimeter information was available. Although insufficient for full scale validation, results provide valuable insights into the strengths and shortcomings of the Global Fire Atlas estimates of individual fire size, duration and expansion rates. In addition, we compared day-to-day expansion rates ($km^2\ day^{-1}$) of individual large fires across both datasets. If multiple

Global Fire Atlas perimeters overlapped with a single US Forest Service fire perimeter, we compared the fires with the largest overlapping surface area."

Line 346: "In line with these findings, we found good agreement between a 3-day running average of Global Fire Atlas and US Forest service estimates of daily fire expansion, but reduced correspondence for daily estimates of fire growth rates due to uncertainty in the day-of-burn of the burned area product (Fig. B1)."

Line 392: "The comparison of Global Fire Atlas data to a small dataset (n = 15) of daily perimeters of large wildfires in primarily forested cover types mapped by the US Forest Service yielded good correspondence between estimates of fire size, duration, and expansion rates (Fig. 7). The improved comparison of fire size (cf. Fig. 5a and 7a) could be related to the US Forest Service data being more accurate than MTBS, but likely also represents the good performance of the Global Fire Atlas (e.g. compare Figs. 7a, b and c to Figs. 7d, e and f) and underlying burned area products (Fusco et al., 2019) for relatively large fires. In contrast to the suggested underestimate of fire duration shown in Fig. 6a, these data suggest the Global Fire Atlas may slightly overestimate fire duration. This difference may reflect the fact that active fire detections may be triggered by smoldering while the burned area product will only register the initial changes in surface reflectance from fire. Based on a small underestimate of overall burned area and overestimate of fire duration by the Global Fire Atlas, the average daily fire expansion rates based on US Forest Service data were higher than estimates based on Global Fire Atlas data (Fig. 7c and f)."

Line 583: "Moreover, the uncertainty in the burn date of the underlying burned area product is typically at least one day, resulting in a large uncertainty in the fire duration estimates of shorter fires. Global Fire Atlas data therefore performed best for large fires (Figs. 6 and 7)."

[Figure]

Figure 1 (Fig. B2 in updated manuscript): **Comparison of Global Fire Atlas perimeters and ignition locations to estimates based on MTBS and VIIRS for frequently-burning grasslands in Kansas, USA.** (a) Global Fire Atlas adjusted burn dates from MCD64A1, (b) per-pixel comparison of adjusted burn dates used within the Global Fire Atlas (GFA) to the day of the (first) active fire detection from VIIRS, (c) ignition points as estimated by the Global Fire Atlas, (d) manually interpreted ignition locations (red circles) based on VIIRS active fire detections on top of MTBS fire perimeters, (e) individual fires as estimated by the Global Fire Atlas, and (f) the MTBS burned area and individual fires. Here, MCD64A1 data underestimates the total burned area compared to the visual interpretation of Landsat data within the MTBS project, resulting in fragmentation of individual large fires. However, the daily temporal resolution of MODIS imagery allows the Global Fire Atlas to distinguish individual fires and ignition points within larger burn scars that cannot be resolved from infrequent Landsat observations used to delineate fire perimeters within the MTBS project. Broad patterns of ignition locations identified by the Global Fire Atlas are confirmed by manual interpretation of patterns inferred from VIIRS active fire detections (d).

[Figure]

Figure 2 (new Fig. 7): **Comparison of Global Fire Atlas (GFA) and US Forest Service (FS) data for a selected number of large wildfires in the US.** Comparison of (a) fire size, (b) duration, and (c) average daily expansion rate for all fires (N=15), (d, e and f) are like (a, b and c) but for fires smaller than 250 km$^2$ (N=12). Correlation coefficients are provided based on linear regression with (yellow) and without (green) intercept, assuming a non-zero intercept could indicate a structural offset between both datasets.

[Figure]

Figure 3 (new **Figure B1**): **Comparison of daily Global Fire Atlas and US Forest Service data for a selected number of well characterized wildfires in the US.** (a) The accumulated daily fire size (for all fires, N=15) illustrates the ability of the Global Fire Atlas to reproduce individual large fire sizes at any specific day over the fire lifetime (each blue dot indicates the size of a specific fire on a specific day). (b) A 3-day running average of the daily growth or "expansion" of each fire (km$^2$ day$^{-1}$) and (c) the daily expansion on each day of each fire. Figures (d), (e), and (f) are like (a), (b), and (c), but for US Forest Service fire sizes smaller than 500 km$^2$ or expansion rates lower than 250 km$^2$ day$^{-1}$ and corresponding Global Fire Atlas estimates (see intermittent boxes on top-figures).

On the other hand, I doubt about the utility of providing global averages of different fire parameters, such as fire duration or progression by continent. In this regard, some of the comments included in the results section may seem quite trivial or difficult to justify empirically. What is the point of concluding that "fire duration exerted a strong control on fire size and total burned area"? Is this not the case in the vast majority of fires?

Although this may seem trivial, the vast majority of fire models currently do not include multi-day fires (e.g. Hantson et al., 2016; Rabin et al., 2017). Our study now for the first time shows that multi-day fires are the norm across all ecosystems and in some ecosystems "duration" exerts a strong control on eventual fire size and total burned area while fire speed is more important in other ecosystems. Incorporating these mechanisms into fire-enabled global ecosystem models is thus critical to capture the (changing) role of fire in the Earth system. We think it is exciting that with these new data we are now for the first time able to analyze how fire behavior influences fire size distributions and eventual burned area. We believe that summarizing these data across continental or ecosystem scales provides a good lookup table for e.g. fire modelers to see whether their model results are within the right range.

In summary, the authors should make an additional effort to really validate their product and better identify the weaknesses of current analysis.

We very much appreciate the suggestion of the reviewer that additional validation data would be helpful, but these data are unfortunately not as readily accessible as the reviewer suggests. In response to this suggestion we have manually compiled a small dataset of well characterized daily behavior of forest fires in the US. Results clearly demonstrate the ability of the Global Fire Atlas to assess individual fire behavior but also illustrate some of the specific shortcomings that we now discuss in more detail. During the coming years we are very much looking forward to further develop our data product as well as provide improved validation and optimization of parameters based on new data availability.

Specific comments

Line 45: Worldwide, fires burn an area larger than the size of the European Union every year (Randerson et al.,2012; Giglio et al., 2013). Please include total area in km2, the reader does not need to know the size of the European union to understand your sentence.

We believe the reader will understand this sentence without knowing the exact size of the European Union as we simply mean "a large area".

Line 55: you claim that burned area reduction is occurring in the last two decades, but Andela et al., 2017 paper refers only to the 2001-2017 period (1995-2001 with more uncertainty), so you could only claim that the reduction is observed in the last few years, as you do not have date from several decades ago.

The study of Andela et al. (2017) included 18 years of data, we will change "Over the past two decades, .." to "Over the past 18 years, .."

Line 65: Our understanding of global fire activity is also severely constrained by the coarse resolution data we are based on our analysis. Recent analysis of burned area estimation comparing coarse and medium resolution data shows that in fact we may be losing a significant part of fire activity (Roteta et al., 2018, https://geogra.uah.es/fire_cci/sfd.php), particularly in tropical regions.

We appreciate the importance of small fires (e.g. Randerson et al., 2012), and we will more clearly discuss the advantages and limitations of the different datasets in our manuscript (see also updated Fig. B2 (Fig. 1 here) and corresponding discussion above). However, we would like to keep our introduction focused on characterizing global fire behavior instead of other important issues that we do not contribute to in this work.

Specifically, we will update line 568: "In line with previous studies, we found that the coarser resolution (500 m) of the MODIS burned area data used to develop the Global Fire Atlas sometimes underestimated overall burned area (e.g. Randerson et al., 2012; Roteta et al., 2019), fragmenting individual large fires. However, the Landsat-based MTBS data at 30 m resolution were unable to distinguish individual fires within large burn patches of fast-moving grassland fires based on infrequent Landsat satellite overpasses (Fig. B2)."

Line 88: update (Giglio et al., submitted)

Done

Lines 155-164: How did you proceed in the case of small fires (a few pixels)? You claim that local minima are deleted when they do not spread forward in time.

In case there is no "later burn date", the ignition point(s) associated with the largest possible number of iterations were retained. We will clarify this in the text.

Line 160: "For short duration fires, the ignition points were retained associated with largest possible number of iterations."

Lines 180-187: Fire spread is obviously associated to wind speed and slope, not just to fuel availability. Therefore the assumptions made by the authors seem quite arbitrary for a global product. Have they made any validation of their persistence algorithm? It is not clear what happened with areas that burned 2 times, were they assigned 6 or 8 day persistency? The thresholds are in fact overlapped.

Our "fire persistence threshold" is somewhat similar to the "cut off" value previously used in flood fill based approaches (e.g. Archibald and Roy, 2009; Hantson et al., 2015; Nogueira et al., 2016; Oom et al., 2016; Laurent et al., 2018). However, in contrast to the flood fill based algorithms, we force the fires to only move forward in time (i.e. logical progression), which can be done because we first apply the ignition point filter that removes small inconsistencies in the burn date estimates. Our threshold values (i.e. 4, 6, 8, or 10 days) were mostly based on the idea that if fire frequencies are low, the probability of multiple fires occurring in each other's vicinity is likely low, hence we can use a longer threshold. In areas of frequent (human caused) fires on the other hand, it is not unlikely to have a new ignition point in the vicinity of a burn scar from a previous fire, in this case we use a short threshold to reduce the likelihood of independent fires to be merged artificially. Fire frequency is also closely related to vegetation patterns, hence we notice that our thresholds are broadly biome dependent (e.g. typically 10-day thresholds in high fuel load boreal and temperate zones and low 4-day thresholds in frequently burning savannas and grasslands).

Following the reviewer's suggestion, we propose to make the following textual clarifications:

Line 185: We will change line 185 to ".., and a 6, 8 and 10-day fire persistence period for grid cells that burned 3 times, 2 times, or 1 time, respectively." to be more precise.

Line 560: "Interestingly, we found similar spatial patterns of fire size (cf. Fig. 8 and Archibald et al., 2013; Hantson et al., 2015), although absolute estimates may show large differences based on the "cut off" value used within the flood-fill approach (Oom et al., 2016), and to a lesser extent by the fire persistence threshold used here."

Line 195. It is not clear if two active fires that merged were assigned a single perimeter or two. It seems they were divided, but most forest services would probably consider them as single one.

We define a single fire as having one ignition point, so several fires that merge would be considered independent fire events in our dataset. This is indeed one of the reasons that our data deviate from the MTBS (also see our response to your earlier suggestions). This is explained in more detail in section 2.1.

Lines 240-: : : It is not clear what the authors did when areas were not observed by clouds or cloud shadows. What is the impact of unobserved periods in fire progression? Were the geometrical deformation effects caused by off-nadir observations taken into account?

We use the MCD64A1 burned area product without any further modification, therefore the uncertainty in the day-of-burn would likely increase during periods of cloud cover (we also mention this, e.g. lines 171-173). Similarly, the scan angle of MODIS instruments (or data-gaps) could potentially affect the correct attribution of burned area to a given day. In fact, this is the reason we let our time series start in 2003, when the combination of the MODIS instruments aboard both Terra and Aqua provide more frequent

observations (see lines 89-90). Nevertheless, the uncertainty in burn date will affect Global Fire Atlas fire characterization, in particular of small and short fires. For example, a multi-pixel single day fire could easily get a longer fire duration assigned solely based on the uncertainty of the burn date in the burned area product. For large multi-day fires, these effects become smaller (e.g. Figs. 4 and 6 of manuscript). Based on the additional comparison of the Global Fire Atlas and US Forest Service data we will more clearly discuss the consequences of uncertainties in the burn date:

Line 340 "Several factors may account for the positive bias in the 500 m day of burn from burned area compared to active fire detections, including orbital coverage, cloud and smoke obscuration, and different thresholds between burned area and active fire algorithms regarding the burnt fraction of a 500 m grid cell."

Line 346 "). In line with these findings, we found good agreement between a 3-day running average of Global Fire Atlas and US Forest service estimates of daily fire expansion, but reduced correspondence for daily estimates of fire growth rates due to uncertainty in the day-of-burn of the burned area product (Fig. B1)."

Figure 3 shows direction of spread that are not very realistic, as all sort of directions are included, even for neighbor pixels (North and South directions in contiguous areas??)

The reviewer should remember that a single pixel represents 21 ha, and may contain numerous landscape features that form natural barriers to fire and could change the fire direction (e.g. vegetation patterns, gullies etc.). Nevertheless, it is true that on a per-pixel level the direction estimate may be quite uncertain, this figure mostly serves to demonstrate how the algorithm works (i.e., for each pixel between fire lines it is estimated how the fire has moved, which results in a speed and direction of spread). Because of the uncertainty at the individual pixel level (e.g. see Fig. 4), we report dominant direction including only multi-day fires larger than 10 km$^2$ in our global map (Fig. 9).

It is not clear why did you include MCD64 in Figure 4, as the date information should be the almost the same as the Global Fire Atlas. I would recommend changing it to a single graph showing dating accuracy for the four major biomes.

We include the MCD64A1 col. 6 data to demonstrate that despite the filters we apply, the overall adjustment of the burn date by the Global Fire Atlas algorithm was small.

Lines 343-346: "The adjustments made to the burn date here, required to effectively determine the extent and duration of individual fires, had a relatively small effect on the overall accuracy but tended to reduce the negative bias in burn dates and increase the positive bias (i.e. delayed burn date compared to active fire detection, see red and black lines in Fig 4)."

The fire dominant direction will probably be more useful for fire modelers expressed in degrees.

Converting the dominant direction to degrees can be achieved by multiplying the numerical dominant direction (ranging from 0-8) by 45. We will include this suggestion in the online user guide.

Other authors have done similar analysis, a recent one by Laurent et al., 2018. Line 440. I doubt that any fire behavior modeler would agree with: "the dominant direction typically represented less than half of the pixels". I think the approach by Laurent et al (2018) using the dominant direction of the evolving ellipsis is more adequate in this regard, as most fires have a dominant wind direction.

We appreciate that fire direction may be estimated in various ways, with likely similar outcomes. We have chosen for the approach we present in our manuscript because it is "internally consistent", in other words, fire direction and speed are derived at the same time when we calculate the most logical (i.e. shortest distance) path the fire may have followed. The exciting thing about the Global Fire Atlas and similar datasets is that, based on the characterization of about one million individual fires worldwide each year, we can now actually investigate what "most" fires do. Our first results indicate that, although dominant wind direction was important, landscape features may be more important than previously thought.

I do not understand the meaning of using average NDVI values to show extreme fires. I do not see the relation.

The NDVI map on the background provides the reader an idea of vegetation cover and available fuels, closely related to fire occurrence and behavior (e.g. Bowman et al., 2009).

We have now clarified this "The background image depicts mean MODIS normalized difference vegetation index (NDVI, 2003 – 2016), an indicator for large scale vegetation patterns and available fuels."

**References**

[revised manuscript text omitted]

---

## Referee Report (RR1)

The Global Fire Atlas of individual fire size, duration, speed, and direction

I thank the authors for the detailed response to my previous comments, although I still disagree about some, and about including a new dataset to validate at least partially their results. As said in the previous version, I appreciate the effort and interest of this work and their relevance for improving fire models at regional and global scales.

One of my main comments of the previous version was related to the shortcomings of the input dataset used by the authors. I realize that MCD64 is probably the best and most used global BA product, and that a real alternative for the authors' goal is not available yet. However, the MCD64 was derived to be a global product, providing reasonable good data at global or continental scales. When using these data for specific fire events, a clear recognition of product limitations should be done. For instance, the authors of the MCD64 algorithm indicated that the product has increased total BA by 26% from c5 to c6 (Giglio et al., 2018). Now, a manuscript has been sent to RSE with a full statistical validation of MCD64 that estimates 40.2% commission error and 72.6% omission error. It is not indicated there if the omissions are caused by small fires or by missing large ones, but the most likely are the former. Roteta et al., 2019, comparison between Sentinel-2 and MCD64 in Africa estimated 80% more BA in the former, while MCD64 did not provide any reliable estimation for fires below 100 ha. All these shortcomings should be reflected in the manuscript.

On the other hand, the authors claim that their algorithm can be used with other datasets, but their manuscript is not about an algorithm, but rather about a product (at least, this is how it is currently written), so the limitations of the product should be indicated to potential users. I am not criticizing the algorithm or the interest of the analysis done, but rather the use of a global product for local analysis without taken into account its actual limitations.

The papers referenced by the authors to support their view (Archibald and Roy, 2009; Hantson et al., 2015; Frantz et al., 2016; Nogueira et al., 2017; Laurent et al., 2018), compute fire metrics from single fires, but only to present them at regional or continental levels, not at the detail of single fires, as it is the case here.

I appreciate the inclusion of comparison with fire behavior information derived from the USFS. I realize the difficulty of getting this information, but these fires are quite particular, as they are quite large and occurring in temperate forest. I wonder if information from Australian or Canadian fires could also be obtained to include a few examples of Tropical and Boreal burns. In addition, figure 7 shows good correlation, but systematic bias for some variables that should also be properly acknowledged.

In summary, I think the authors should include a more detailed discussion on the strengths and limits of their product, considering the actual limitations of the input dataset, which in my view was never developed to derived single fire information. Considering the MCD64 product misses 72% of burned pixels globally, following recent validation estimates from the actual authors of the MCD64 product, the potential user should at least use the GFA product with caution.

Specific comments

Line 45: Worldwide, fires burn an area larger than the size of the European Union every year (Randerson et al.,2012; Giglio et al., 2013). Please include total area in km2, the reader does not need to know the size of the European union to understand your sentence.

We believe the reader will understand this sentence without knowing the exact size of the European Union as we simply mean "a large area".

I do not see your point. Why you should not just include the total BA in km2 and then compare to whatever territory you consider most adequate? In addition, your comparison is not very precise, as the European Union will have 250.000 km2 less in a few weeks.

Line 109: Several studies have shown that the MCD64A1 col. 6 burned area product provides a considerable improvement compared to previous generation of moderate resolution global burned area products (Padilla et al., 2015; Giglio et al., 2018; Humber et al., 2018).

Please note that the first study compared then existing BA products with c5, not with c6, and that Giglio et al., 2018 compared it with previous c5, not with other products.

Line 274: I appreciate the effort and the difficulties to validate the product, but what the authors present is not really a validation, just a preliminary assessment. A real validation would imply a statistically design sample of their product against reference data. I would use another title for this section.

Line 429: R2 values in Figure 7 should be complemented by RMS. As the figure shows, even the highest r2 may have important bias.

Line 629: "Reasonable correlations ($r_2$ ranging from 0.3 to 0.5) were found between Global Fire Atlas". I see the authors still consider "reasonable" correlations than explain less than 50% of the variance. I am not sure what reasonable means in this case, but statistically I think they are not very encouraging.

---

## Author Response (AR2)

**The Global Fire Atlas of individual fire size, duration, speed, and direction**

I thank the authors for the detailed response to my previous comments, although I still disagree about some, and about including a new dataset to validate at least partially their results. As said in the previous version, I appreciate the effort and interest of this work and their relevance for improving fire models at regional and global scales.

One of my main comments of the previous version was related to the shortcomings of the input dataset used by the authors. I realize that MCD64 is probably the best and most used global BA product, and that a real alternative for the authors' goal is not available yet. However, the MCD64 was derived to be a global product, providing reasonable good data at global or continental scales. When using these data for specific fire events, a clear recognition of product limitations should be done. For instance, the authors of the MCD64 algorithm indicated that the product has increased total BA by 26% from c5 to c6 (Giglio et al., 2018). Now, a manuscript has been sent to RSE with a full statistical validation of MCD64 that estimates 40.2% commission error and 72.6% omission error. It is not indicated there if the omissions are caused by small fires or by missing large ones, but the most likely are the former. Roteta et al., 2019, comparison between Sentinel-2 and MCD64 in Africa estimated 80% more BA in the former, while MCD64 did not provide any reliable estimation for fires below 100 ha. All these shortcomings should be reflected in the manuscript.

On the other hand, the authors claim that their algorithm can be used with other datasets, but their manuscript is not about an algorithm, but rather about a product (at least, this is how it is currently written), so the limitations of the product should be indicated to potential users. I am not criticizing the algorithm or the interest of the analysis done, but rather the use of a global product for local analysis without taken into account its actual limitations.

The papers referenced by the authors to support their view (Archibald and Roy, 2009; Hantson et al., 2015; Frantz et al., 2016; Nogueira et al., 2017; Laurent et al., 2018), compute fire metrics from single fires, but only to present them at regional or continental levels, not at the detail of single fires, as it is the case here.

I appreciate the inclusion of comparison with fire behavior information derived from the USFS. I realize the difficulty of getting this information, but these fires are quite particular, as they are quite large and occurring in temperate forest. I wonder if information from Australian or Canadian fires could also be obtained to include a few examples of Tropical and Boreal burns. In addition, figure 7 shows good correlation, but systematic bias for some variables that should also be properly acknowledged.

In summary, I think the authors should include a more detailed discussion on the strengths and limits of their product, considering the actual limitations of the input dataset, which in my view was never developed to derived single fire information. Considering the MCD64 product misses 72% of burned pixels globally, following recent validation estimates from the actual authors of the MCD64 product, the potential user should at least use the GFA product with caution.

We thank the reviewer for these suggestions to further clarify the strengths and limitations of the Global Fire Atlas, derived from the MCD64A1 Collection 6 global burned area product. In

response to these suggestions, and a similar request from the Editor, we now end our discussion with a specific paragraph that recognizes the limitations of moderate resolution burned area products for estimates of individual fire behaviour and recognizes the potential to improve the Global Fire Atlas through the use of higher resolution burned area information:

"In addition to the Global Fire Atlas algorithm, the data quality also depends on the underlying global burned area product (MCD64A1 c6). In particular, several recent studies have shown that moderate resolution burned area products are unable to adequately map the occurrence of small fires ($\sim \leq 100$ ha) in the United States (Fusco et al., 2019) and savanna regions of Brazil (Rodrigues et al., 2019) and Africa (Roteta et al., 2019), resulting in a considerable underestimate of global burned area (Randerson et al., 2012; Giglio et al., 2018). Therefore, care should be taken when using the Global Fire Atlas for cropland regions or other regions dominated by small fires (see Fig. 8c). The quality of derived parameters in the Global Fire Atlas for these same regions also depends on the fire persistence threshold we used to identify when fires spread from one grid cell into the next. The thresholds we used may be more appropriate for analysis of fires in natural landscapes than in croplands with synchronized small fire activity across multiple adjacent fields. Finally, daily burned area products do not resolve the diurnal cycle of fire activity; fire lifetime and fire behavior may vary widely across fire regimes (Freeborn et al., 2011; Andela et al., 2015), and sub-daily fire dynamics cannot be resolved in the Global Fire Atlas. In line with these limitations, we found that Global Fire Atlas data performed best for large fires (Figs. 6 and 7). Further development of the Fire Atlas product suite is possible based on improvements in the underlying burned area data from multiple satellite sensors as well as new active fire products at higher spatial resolution (e.g., VIIRS). The Global Fire Atlas algorithm provides a flexible framework that can be easily adjusted to work at different spatial or temporal resolutions."

Concerning the specific suggestions following the inclusion of USFS daily fire behavior estimates, we very much appreciate this suggestion but we are unfortunately not aware of any existing data that can be easily obtained for comparison. We are currently working on a separate manuscript that explores the use of higher resolution (375m) VIIRS active fire data to track (sub)daily progression of wildfires in the US and we think that this may provide useful data for further validation and/or an improved version of the Global Fire Atlas in due course. In response to this suggestion and the specific comment below, we will change the term "validation" to "preliminary accuracy assessment". We will also more specifically discuss the structural biases observed for fire perimeters and duration (Figs. 5 and 7):

"Overall, the Global Fire Atlas underestimated fire perimeter length in all of the vegetation classes."

"Both comparisons (Figs. 6, 7b and 7e) suggest the Global Fire Atlas may overestimate the duration of smaller fires with relatively short duration, likely based on the uncertainty in underlying burn dates."

**Specific comments**

Line 45: Worldwide, fires burn an area larger than the size of the European Union every year (Randerson et al.,2012; Giglio et al., 2013). Please include total area in km2, the reader does not need to know the size of the European Union to understand your sentence.
We believe the reader will understand this sentence without knowing the exact size of the European Union as we simply mean "a large area".

I do not see your point. Why you should not just include the total BA in km2 and then compare to whatever territory you consider most adequate? In addition, your comparison is not very precise, as the European Union will have 250.000 km2 less in a few weeks.

95

We have now further clarified this and changed the sentence to:

"Worldwide, fires burn an area about the size of the European Union every year (423 Mha yr-1; Giglio et al., 2018)."

100

Line 109: Several studies have shown that the MCD64A1 col. 6 burned area product provides a considerable improvement compared to previous generation of moderate resolution global burned area products (Padilla et al., 2015; Giglio et al., 2018; Humber et al., 2018). Please note that the first study compared then existing BA products with c5, not with c6, and that Giglio et al., 2018 compared it with previous c5, not with other products.

105

We appreciate that Padilla et al. (2015) compare different data-sets to MCD64A1 c5 and not c6, and will therefore replace "Padilla et al. (2015)" in this sentence with "Rodrigues et al. (2019)". Rodrigues et al. (2019) compare burned area estimates based on Landsat to MCD64A1 c5 and c6 for the Brazilian Cerrado.

110

Line 274: I appreciate the effort and the difficulties to validate the product, but what the authors present is not really a validation, just a preliminary assessment. A real validation would imply a statistically design sample of their product against reference data. I would use another title for this section.

We acknowledge the reviewer's concern and have changed the title of the section in question to

115

"Preliminary accuracy assessment" accordingly. We note, however, that real validation does not in fact require a statistically designed sample as claimed by the reviewer. The CEOS Land Product validation protocol (Morisette et al., 2006) includes a statistically based sampling strategy only at Stage 3 and above; Stages 1 and 2 in the CEOS validation hierarchy have no such requirement.

120

Line 429: R2 values in Figure 7 should be complemented by RMS. As the figure shows, even the highest r2 may have important bias.

We thank the reviewer for this suggestion and now note the root-mean-square deviation on each of the figure panels.

125

Line 629: "Reasonable correlations (r2 ranging from 0.3 to 0.5) were found between Global Fire Atlas". I see the authors still consider "reasonable" correlations than explain less than 50% of the variance. I am not sure what reasonable means in this case, but statistically I think they are not very encouraging.

We have now changed the sentence to: "Low to moderate correlations ($r^2$ ranging from 0.3 to 0.5)

130

were found between Global Fire Atlas and fire duration estimates based on a combination of MTBS fire perimeters and VIIRS active fire detections."

[revised manuscript text omitted]